## SCIENCE FORUM

# Ten common statistical mistakes to watch out for when writing or reviewing a manuscript

**Abstract** Inspired by broader efforts to make the conclusions of scientific research more robust, we have compiled a list of some of the most common statistical mistakes that appear in the scientific literature. The mistakes have their origins in ineffective experimental designs, inappropriate analyses and/or flawed reasoning. We provide advice on how authors, reviewers and readers can identify and resolve these mistakes and, we hope, avoid them in the future.

**TAMAR R MAKIN\* AND JEAN-JACQUES ORBAN DE XIVRY**

Much has been written about the need to improve the reproducibility of research (*Bishop, 2019*; *Munafò et al., 2017*; *Open Science Collaboration, 2015*; *Weissgerber et al., 2018*), and there have been many calls for improved training in statistical analysis techniques (*Schroter et al., 2008*). In this article we discuss ten statistical mistakes that are commonly found in the scientific literature. Although many researchers have highlighted the importance of transparency and research ethics (*Baker, 2016*; *Nosek et al., 2015*), here we discuss statistical oversights which are out there in plain sight in papers that advance claims that do not follow from the data – papers that are often taken at face value despite being wrong (*Harper and Palayew, 2019*; *Nissen et al., 2016*; *De Camargo, 2012*). In our view, the most appropriate checkpoint to prevent erroneous results from being published is the peer-review process at journals, or the online discussions that can follow the publication of preprints. The primary purpose of this commentary is to provide reviewers with a tool to help identify and manage these common issues.

All of these mistakes are well known and there have been many articles written about them, but they continue to appear in journals. Previous commentaries on this topic have tended to focus on one mistake, or several related mistakes: by discussing ten of the most common mistakes we hope to provide a resource that researchers can use when reviewing manuscripts or commenting on preprints and published papers. These guidelines are also intended to be useful for researchers planning experiments, analysing data and writing manuscripts.

Our list has its origins in the journal club at the London Plasticity Lab, which discusses papers in neuroscience, psychology, clinical and bioengineering journals. It has been further validated by our experiences as readers, reviewers and editors. Although this list has been inspired by papers relating to neuroscience, the relatively simple issues described here are relevant to any scientific discipline that uses statistics to assess findings. For each common mistake in our list we discuss how the mistake can arise, explain how it can be detected by authors and/or referees, and offer a solution.

We note that these mistakes are often interdependent, such that one mistake will likely impact others, which means that many of them

**Figure 1.** Interpreting comparisons between two effects without directly comparing them. (A) Two variables, *X* and *Y*, were measured for two groups A and B. It looks clear that the correlation between these two variables does not differ across these two groups. However, if one compares both correlation coefficients to zero by calculating the significance of the Pearson's correlation coefficient *r*, it is possible to find that one group (group A; black circles; *n* = 20) has a statistically significant correlation (based on a threshold of p≤0.05), whereas the other group (group B, red circles; *n* = 20) does not. However, this does not indicate that the correlation between the variables X and Y differs between these groups. Monte Carlo simulations can be used to compare the correlations in the two groups (*Wilcox and Tian, 2008*). (B) In another experimental context, one can look at how a specific outcome measure (e.g. the difference pre- and post-training) differs between two groups. The means for groups C and D are the same, but the variance for group D is higher. If one uses a one-sample t-test to compare this outcome measure to zero for each group separately, it is possible to find that, this variable is significantly different from zero for one group (group C; left; *n* = 20), but not for the other group (group D, right; *n* = 20). However, this does not inform us whether this outcome measure is different between the two groups. Instead, one should directly compare the two groups by using an unpaired t-test (top): this shows that this outcome measure is not different for the two groups. Code (including the simulated data) available at github.com/jjodx/InferentialMistakes (*Makin and Orban de Xivry, 2019*; https://github.com/elifesciences-publications/InferentialMistakes).
DOI: https://doi.org/10.7554/eLife.48175.002

cannot be remedied in isolation. Moreover, there is usually more than one way to solve each of these mistakes: for example, we focus on frequentist parametric statistics in our solutions, but there are often Bayesian solutions that we do not discuss (*Dienes, 2011*; *Etz and Vandekerckhove, 2016*).

To promote further discussion of these issues, and to consolidate advice on how to best solve them, we encourage readers to offer alternative solutions to ours by annotating the online version of this article (by clicking on the 'annotations' icon). This will allow other readers to benefit from a diversity of ideas and perspectives.

We hope that greater awareness of these common mistakes will help make authors and reviewers more vigilant in the future so that the mistakes become less common.

## Absence of an adequate control condition/group

### The problem
Measuring an outcome at multiple time points is a pervasive method in science in order to assess the effect of an intervention. For instance, when examining the effect of training, it is common to probe changes in behaviour or a physiological measure. Yet, changes in outcome measures can arise due to other elements of the study that do not directly relate to the manipulation (e.g. training) per se. Repeating the same task in the absence of an intervention might induce a change in the outcomes between pre- and post-intervention measurements, e.g. due to the participant or the experimenter merely becoming accustomed to the experimental setting, or due to other changes relating to the passage of time. Therefore, for any studies looking at the

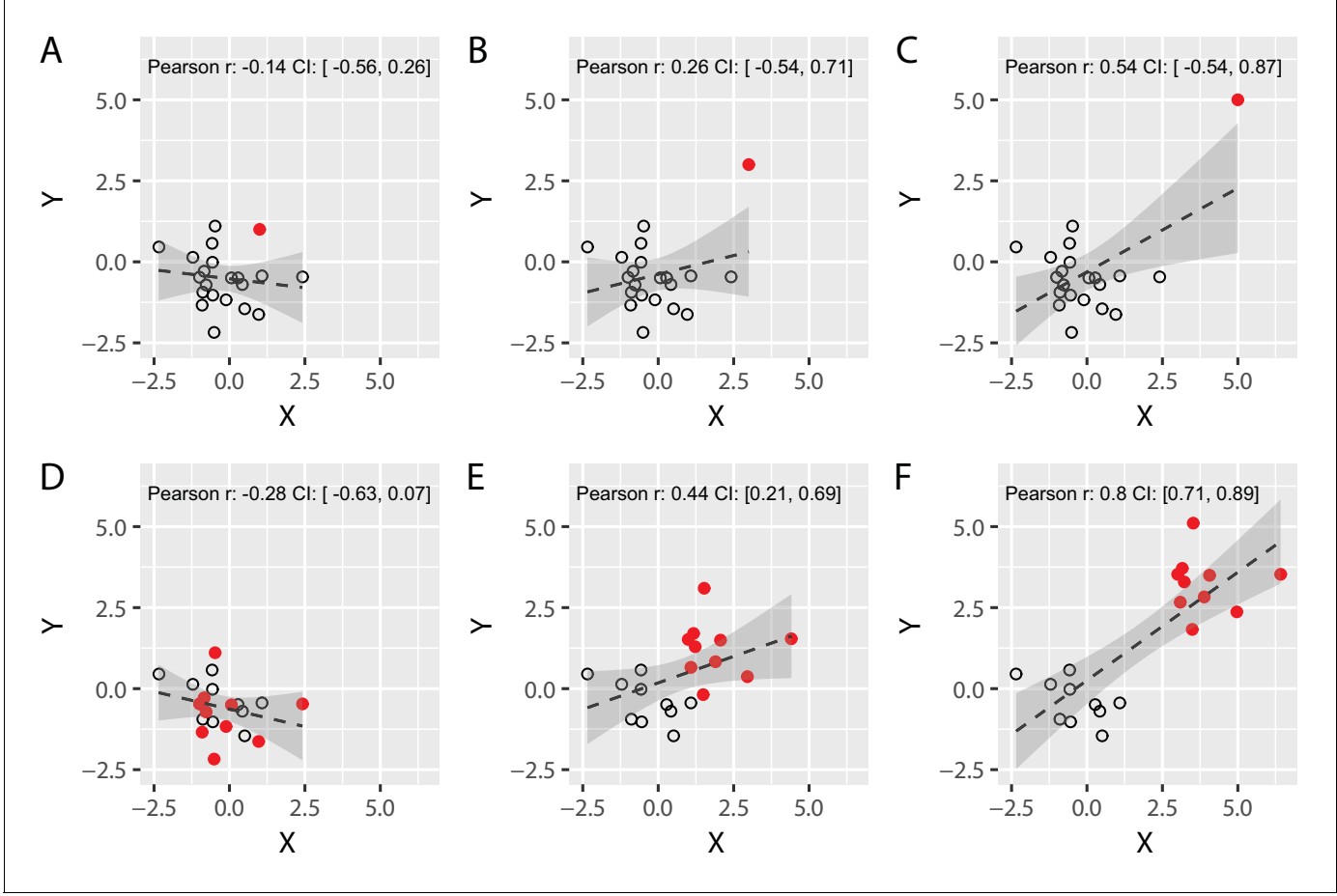

**Figure 2.** Spurious correlations: the effect of a single outlier and of subgroups on Pearson's correlation coefficients. (A–C) We simulated two different uncorrelated variables with 19 samples (black circles) and added an additional data point (solid red circle) whose distance from the main population was systematically varied until it became a formal outlier (panel C). Note that the value of Pearson's correlation coefficient R artificially increases as the distance between the main population and the red data point is increased, demonstrating that a single data point can lead to spurious Pearson's correlations. (D–F) We simulated two different uncorrelated variables with 20 sample that were arbitrarily divided into two subgroups (red vs. black, N = 10 each). We systematically varied the distance between the two subgroups from panel D to panel F. Again, the value of R artificially increases as the distance between the subgroups is increased. This shows that correlating variables without taking the existence of subgroups into account can yield spurious correlations. Confidence intervals (CI) are shown in grey, and were obtained via a bootstrap procedure (with the grey region representing the region between the 2.5 and 97.5 percentiles of the obtained distribution of correlation values). Code (including the simulated data) available at github.com/jjodx/InferentialMistakes.

DOI: https://doi.org/10.7554/eLife.48175.003

effect of an experimental manipulation on a variable over time, it is crucial to compare the effect of this experimental manipulation with the effect of a control manipulation.

Sometimes a control group or condition is included, but is designed or implemented inadequately, by not including key factors that could impact the tracked variable. For example, the control group often does not receive a 'sham' intervention, or the experimenters are not blinded to the expected outcome of the intervention, contributing to inflated effect sizes (*Holman et al., 2015*). Other common biases

result from running a small control group that is insufficiently powered to detect the tracked change (see below), or a control group with a different baseline measure, potentially driving spurious interactions (*Van Breukelen, 2006*). It is also important that the control and experimental groups are sampled at the same time and with randomised allocation, to minimise any biases. Ideally, the controlled manipulation should be otherwise identical to the experimental manipulation in terms of design and statistical power and only differ in the specific stimulus dimension or variable under manipulation. In

doing so, researchers will ensure that the effect of the manipulation on the tracked variable is larger than variability over time that is not directly driven by the desired manipulation. Therefore, reviewers should always request for controls in situations where a variable is compared over time.

### How to detect it

Conclusions are drawn on the basis of data of a single group, with no adequate control conditions. The control condition/group does not account for key features of the task that are inherent to the manipulation.

### Solutions for researchers

If the experimental design does not allow for separating the effect of time from the effect of the intervention, then conclusions regarding the impact of the intervention should be presented as tentative.

Further reading
(*Knapp, 2016*).

## Interpreting comparisons between two effects without directly comparing them

### The problem

Researchers often base their conclusions regarding the impact of an intervention (such as a pre- vs. post-intervention difference or a correlation between two variables) by noting that the intervention yields a significant effect in the experimental condition or group, whereas the corresponding effect in the control condition or group is not significant. Based on these two separate test outcomes, researchers will sometimes suggest that the effect in the experimental condition or group is larger than the effect in the control condition. This type of erroneous inference is very common but incorrect. For instance, as illustrated in *Figure 1A*, two variables $X$ and $Y$, each measured in two different groups of 20 participants, could have different outcomes in terms of statistical significance: a correlation co-efficient for the correlation between the two variables in group A might be statistically significant (ie, have $p \leq 0.05$), whereas a similar correlation co-efficient might not be statistically significant for group B. This could happen even if the relationship between the two variables is virtually identical for the two groups (*Figure 1A*), so one

should *not* infer that one correlation is greater than the other.

A similar issue occurs when estimating the effect of an intervention measured in two different groups: the intervention could yield a significant effect in one group but not in the other (*Figure 1B*). Again, however, this does not mean that the effect of the intervention is different between the two groups; indeed in this case, the two groups do not significantly differ. One can only conclude that the effect of an intervention is different from the effect of a control intervention through a direct statistical comparison between the two effects. Therefore, rather than running two separate tests, it is essential to use one statistical test to compare the two effects.

### How to detect it

This problem arises when a conclusion is drawn regarding a difference between two effects without statistically comparing them. This problem can occur in any situation where researchers make an inference without performing the necessary statistical analysis.

### Solutions for researchers

Researchers should compare groups directly when they want to contrast them (and reviewers should point authors to *Nieuwenhuis et al., 2011* for a clear explanation of the problem and its impact). The correlations in the two groups can be compared with Monte Carlo simulations (*Wilcox and Tian, 2008*). For group comparisons, ANOVA might be suitable. Although nonparametric statistics offers some tools (e.g., *Leys and Schumann, 2010*), these require more thought and customisation.

Further reading
(*Nieuwenhuis et al., 2011*).

## Inflating the units of analysis

### The problem

The experimental unit is the smallest observation that can be randomly and independently assigned, i.e. the number of independent values that are free to vary (*Parsons et al., 2018*). In classical statistics, this unit will reflect the degrees of freedom (*df*): For example, when inferring group results, the experimental unit is the number of subjects tested, rather than the number of observations made within each subject. But unfortunately, researchers tend to mix up these measures, resulting in both conceptual

and practical issues. Conceptually, without clear identification of the appropriate unit to assess variation that sub-serves the phenomenon, the statistical inference is flawed. Practically, this results in a spuriously higher number of experimental units (e.g., the number of observations across all subjects is usually greater than the number of subjects). When *df* increases, the critical statistical threshold against which statistical significance is judged decreases, making it easier to observe a significant result if there is a genuine effect (increase of statistical power). This is because there is greater confidence in the outcome of the test.

To illustrate this issue, let us consider a simple pre-post longitudinal design for an intervention study in 10 participants where the researchers are interested in evaluating whether there is a correlation between their main measure and a clinical condition using a simple regression analysis. Their unit of analysis should be the number of data points (1 per participant, 10 in total), resulting in 8 *df*. For *df* = 8, the critical *R* value (with an alpha level of. 05) for achieving significance is 0.63. That is, any correlation above the critical value will be significant ($p \leq 0.05$). If the researchers combine the pre and post measures across participants, they will end up with *df* = 18, the critical *R* value is now 0.44, rendering it easier to observe a statistically significant effect. This is inappropriate because they are mixing within- and between- analysis units, resulting in *dependencies* between their measures – the pre-score of a given subject cannot be varied without impacting their post-score, meaning they only truly have 8 independent *df*. This often results in interpretation of the results as significant when in fact the evidence is insufficient to reject the possibility that there is no effect.

### How detect it
The reviewer should consider the appropriate unit of analysis. If a study aims to understand group effects, then the unit of analysis should reflect the variance across subjects, not within subjects.

### Solutions for researchers
Perhaps the best available solution to this issue is using a mixed-effects linear model, where researchers can define the variability within subjects as a fixed effect, and the between-subject variability as a random effect. This increasingly popular approach (*Boisgontier and Cheval,*

*2016*) allows one to put all the data in the model without violating the assumption of independence. However, it can be easily misused (*Matuschek et al., 2017*) and requires advanced statistical understanding, and as such should be applied and interpreted with some caution. For a simple regression analysis, the researchers have several available solutions to this issue, the easiest of which is to calculate the correlation for each observation separately (e.g. pre, post) and interpret the *R* values based on the existing *df*. The researchers can also average the values across observations, or calculate the correlation for pre/post separately and then average the resulting *R* values (after applying normalisation of the *R* distribution, e.g. *r*-to-*Z* transformation), and interpret them accordingly.

Further reading
(*Pandey and Bright, 2008*; *Parsons et al., 2018*).

## Spurious correlations

### The problem
Correlations are an important tool in science in order to assess the magnitude of an association between two variables. Yet, the use of parametric correlations, such as Pearson's *R* relies on a set of assumptions, which are important to consider as violation of these assumptions may give rise to spurious correlations. Spurious correlations most commonly arise if one or several outliers are present for one of the two variables. As illustrated in the top row of *Figure 2*, a single value away from the rest of the distribution can inflate the correlation coefficient. Spurious correlations can also arise from clusters, e.g. if the data from two groups are pooled together when the two groups differ in those two variables (as illustrated in the bottom row of *Figure 2*).

It is important to note that an outlier might very well provide a genuine observation which obeys the law of the phenomenon that you are trying to discover, in other words – the observation in itself is not necessarily spurious. Therefore, removal of 'extreme' data points should also be considered with great caution. But if this true observation is at risk of violating the assumptions of your statistical test, it becomes spurious *de facto*, and will therefore require a different statistical tool.

### How to detect it

Reviewers should pay particular attention to reported correlations that are not accompanied by a scatterplot and consider if sufficient justification has been provided when data points have been discarded. In addition, reviewers need to make sure that between-group or between-condition differences are taken into account if data are pooled together (see 'Inflating the units of analysis' above).

### Solutions for researchers

Robust correlation methods (e.g. bootstrapping, data winsorizing, skipped correlations) should be preferred in most circumstances because they are less sensitive to outliers (*Salibian-Barrera and Zamar, 2002*). This is because these tests take into consideration the structure of the data (*Wilcox, 2016*). When using parametric statistics, data should be screened for violation of the key assumptions, such as independence of data points, as well as the presence of outliers.

Further reading
(*Rousselet and Pernet, 2012*).

## Use of small samples

### The problem

When a sample size is small, one can only detect large effects, thereby leaving high uncertainty around the estimate of the true effect size and leading to an overestimation of the actual effect size (*Button et al., 2013*). In frequentist statistics in which a significance threshold of *alpha*=0.05 is used, 5% of all statistical tests will yield a significant result in the absence of an actual effect (false positives; Type I error). Yet, researchers are more likely to consider a correlation with a high coefficient (e.g. $R>0.5$) as robust than a modest correlation (e.g. $R=0.2$). With small sample sizes, the effect size of these false positives is large, giving rise to the significance fallacy: "If the effect size is *that* big with a small sample, it can only be true." (This incorrect inference is noted in *Button et al., 2013*). Critically, the larger correlation is not a result of there being a stronger relationship between the two variables, it is simply because the overestimation of the actual correlation coefficient (here, $R = 0$) will always be larger with a small sample size. For instance, when sampling two uncorrelated variables with $N = 15$, simulated false-positive correlations roughly range between |0.5-0.75|

whereas when sampling the same uncorrelated variables with $N = 100$ yields false-positive correlations in the range |0.2-0.25| (Code available at github.com/jjodx/InferentialMistakes).

Designs with a small sample size are also more susceptible to missing an effect that exists in the data (Type II error). For a given effect size (e.g., the difference between two groups), the chances are greater for detecting the effect with a larger sample size (this likelihood is referred to as statistical power). Hence, with large samples, you reduced the likelihood of not detecting an effect when one is actually present.

Another problem related to small sample size is that the distribution of the sample is more likely to deviate from normality, and the limited sample size makes it often impossible to rigorously test the assumption of normality (*Ghasemi and Zahediasl, 2012*). In regression analysis, deviations from the distribution might produce extreme outliers, resulting in spurious significant correlations (see 'Spurious correlations' above).

### How to detect it

Reviewers should critically examine the sample size used in a paper and, judge whether the sample size is sufficient. Extraordinary claims based on a limited number of participants should be flagged in particular.

### Solutions for researchers

A single effect size or a single *p*-value from a small sample is of limited value and reviewers can refer the researchers to *Button et al. (2013)* to make this point. The researchers should either present evidence that they have been sufficiently powered to detect the effect to begin with, such as through the presentation of an a priori statistical power analysis, or perform a replication of their study. The challenge with power calculations is that these should be based on an a priori calculation of effect size from an independent dataset, and these are difficult to assess in a review. Bayesian statistics offer opportunities to determine the power for identifying an effect *post hoc* (*Kruschke, 2011*). In situations where sample size may be inherently limited (e.g. research with rare clinical populations or non-human primates), efforts should be made to provide replications (both within and between cases) and to include sufficient controls (e.g. to establish confidence intervals). Some statistical solutions are offered for assessing case studies (e.g., the Crawford *t*-test; *Corballis, 2009*).

Further reading
(*Button et al., 2013*).

## Circular analysis

### The problem
Circular analysis is any form of analysis that retrospectively selects features of the data to characterise the dependent variables, resulting in a distortion of the resulting statistical test (*Kriegeskorte et al., 2010*). Circular analysis can take many shapes and forms, but it inherently relates to recycling the same data to first characterise the test variables and then to make statistical inferences from them, and is thus often referred to as 'double dipping' (*Kriegeskorte et al., 2009*). Most commonly, circular analysis is used to divide (e.g. subgrouping, binning) or reduce (e.g. defining a region of interest, removing 'outliers') the complete dataset using a selection criterion that is retrospective and inherently relevant to the statistical outcome.

For example, let's consider a study of a neuronal population firing rate in response to a given manipulation. When comparing the population as a whole, no significant differences are found between pre and post manipulation. However, the researchers observe that some of the neurons respond to the manipulation by increasing their firing rate, whereas others decrease in response to the manipulation. They therefore split the population to sub-groups, by binning the data based on the activity levels observed at baseline. This leads to a significant interaction effect – those neurons that initially produced low responses show response increases, whereas the neurons that initially showed relatively increased activity exhibit reduced activity following the manipulation. However, this significant interaction is a result of the distorting selection criterion and a combination of statistical artefacts (regression to the mean, floor/ceiling effects), and could therefore be observed in pure noise (*Holmes, 2009*).

Another common form of circular analysis is when dependencies are created between the dependent and independent variables. Continuing with the example from above, researchers might report a correlation between the cell response post-manipulation and between the difference in cell response across the pre- and post-manipulation. But both variables are highly dependent on the post-manipulation measure. Therefore, neurons that by chance fire more strongly in the post manipulation measure are likely to show greater changes relative to the independent pre-manipulation measure, thus inflating the correlation (*Holmes, 2009*).

Selective analysis is perfectly justifiable when the results are statistically independent of the selection criterion under the null hypothesis. However, circular analysis recruits the noise (inherent to any empirical data) to inflate the statistical outcome, resulting in distorted and hence invalid statistical inference.

### How to detect it
Circular analysis manifests in many different forms, but in principle occurs whenever the statistical test measures are biased by the selection criteria in favour of the hypothesis being tested. In some circumstances this is very clear, e.g. if the analysis is based on data that were selected for showing the effect of interest, or an inherently related effect. In other circumstances the analysis could be convoluted and require more nuanced understanding of co-dependencies across selection and analysis steps (see, for example, Figure 1 in *Kilner, 2013* and the supplementary materials in *Kriegeskorte et al., 2009*). Reviewers should be alerted by impossibly high effect sizes which might not be theoretically plausible, and/or are based on relatively unreliable measures (if two measures have poor internal consistency this limits the potential to identify a meaningful correlation; *Vul et al., 2009*). In that case, the reviewers should ask the authors for a justification for the independence between the selection criteria and the effect of interest.

### Solutions for researchers
Defining the analysis criteria in advance and independently of the data will protect researchers from circular analysis. Alternatively, since circular analysis works by 'recruiting' noise to inflate the desired effect, the most straightforward solution is to use a different dataset (or different part of your dataset) for specifying the parameters for the analysis (e.g. selecting your sub-groups) and for testing your predictions (e.g. examining differences across the sub-groups). This division can be done at the participant level (using a different group to identify the criteria for reducing the data) or at the trial level (using different trials but from all participants). This can be achieved without losing statistical power using bootstrapping approaches (*Curran-Everett, 2009*). If suitable, the reviewer could

ask the authors to run a simulation to demonstrate that the result of interest is not tied to the noise distribution and the selection criteria.

Further reading
(*Kriegeskorte et al., 2009*).

## Flexibility of analysis: *p*-hacking

### The problem

Using flexibility in data analysis (such as switched outcome parameters, adding covariates, undetermined or erratic pre-processing pipeline, *post hoc* outlier or subject exclusion; *Wicherts et al., 2016*) increases the probability of obtaining significant *p*-values (*Simmons et al., 2011*). This is because normative statistics rely on probabilities and therefore the more tests you run the more likely you are to encounter a false positive result. Therefore, observing a significant *p*-value in a given dataset is not necessarily complicated and one can always come up with a plausible explanation for any significant effect particularly in the absence of specific predictions. Yet, the more variation in one's analysis pipeline, the greater the likelihood that observed effects are not genuine. Flexibility in data analysis is especially visible when the same community reports the same outcome variable but computes the value of this variable in different ways across the paper (e.g. www.flexiblemeasures.com; *Carp, 2012*; *Francis, 2013*) or when clinical trials switch their outcomes (*Altman et al., 2017*; *Goldacre et al., 2019*).

This problem can be pre-empted by using standardised analytic approaches, pre-registration of the design and analysis (*Nosek and Lakens, 2014*), or undertaking a replication study (*Button et al., 2013*). Note that pre-registration of experiments can be performed after the results of a first experiment are known and before an internal replication of that effect is sought. But perhaps the best way to prevent *p*-hacking is to show some tolerance to borderline or non-significant results. In other words, if the experiment is well designed, executed, and analysed, reviewers should not 'punish' the researchers for their data.

### How to detect it

Flexibility of analysis is difficult to detect because researchers rarely disclose all the necessary information. In the case of pre-registration or clinical trial registration, the reviewer should compare the analyses performed with the planned analyses. In the absence of pre-registration, it is almost impossible to detect some forms of *p*-hacking. Yet, reviewers can estimate whether all the analysis choices are well justified, whether the same analysis plan was used in previous publications, whether the researchers came up with a questionable new variable, or whether they collected a large battery of measures and only reported a few significant ones. Practical tips for detecting likely positive findings are summarized in *Forstmeier et al. (2017)*.

### Solutions for researchers

Researchers should be transparent in the reporting of the results, e.g. distinguishing pre-planned versus exploratory analyses and predicted versus unexpected results. As we discuss below, exploratory analyses using flexible data analysis are fine if they are reported and interpreted as such in a transparent manner and especially so if they serve as the basis for a replication with pre-specified analyses (*Curran-Everett and Milgrom, 2013*). Such analyses can be a valuable justification for additional research but cannot be the foundation for strong conclusions.

Further reading
(*Kerr, 1998*; *Simmons et al., 2011*).

## Failing to correct for multiple comparisons

### The problem

When researchers explore task effects, they often explore the effect of multiple task conditions on multiple variables (behavioural outcomes, questionnaire items, etc.), sometimes with an underdetermined a priori hypothesis. This practice is termed exploratory analysis, as opposed to confirmatory analysis, which by definition is more restrictive. When performed with frequentist statistics, conducting multiple comparisons during exploratory analysis can have profound consequences for the interpretation of significant findings. In any experimental design involving more than two conditions (or a comparison of two groups), exploratory analysis will involve multiple comparisons and will increase the probability of detecting an effect even if no such effect exists (false positive, type I error). In this case, the larger the number of factors, the greater the number of tests that can be performed. As a result, the probability of observing a false-positive increases (family-wise error rate).

For example, in a $2 \times 3 \times 3$ experimental design the probability of finding at least one significant main or interaction effect is 30%, even when there is no effect (*Cramer et al., 2016*).

This problem is particularly salient when conducting multiple independent comparisons (e.g. neuroimaging analysis, multiple recorded cells or EEG). In such cases, researchers are technically deploying statistical tests within every voxel/cell/timepoint, thereby increasing the likelihood of detecting a false positive result, due to the large number of measures included in the design. For example, Bennett and colleagues (*Bennett et al., 2009*) identified a significant number of active voxels in a dead Atlantic Salmon (activated during a 'mentalising' task) when not correcting for multiple comparisons. This example demonstrates how easy it can be to identify a spurious significant result. Although it is more problematic when the analyses are exploratory, it can still be a concern when a large set of analyses are specified a priori for confirmatory analysis.

### How to detect it

Failing to correct for multiple comparisons can be detected by addressing the number of independent variables measured and the number of analyses performed. If only one of these variables correlated with the dependent variable, then the rest is likely to have been included to increase the chance of obtaining a significant result. Therefore, when conducting exploratory analyses with a large set of variables (such as genes or MRI voxels), it is simply unacceptable for the researchers to interpret results that have not survived correction for multiple comparisons, without clear justification. Even if the researchers offer a rough prediction (e.g. that the effect should be observed in a specific brain area or at an approximate latency), if this prediction could be tested over multiple independent comparisons, it requires correction for multiple comparisons.

### Solutions for researchers

Exploratory testing can be absolutely appropriate, but should be acknowledged. Researchers should disclose all measured variables and properly implement the use of multiple comparison procedures. For example, applying standard corrections for multiple comparisons unsurprisingly resulted in no active voxels in the dead fish example (*Bennett et al., 2009*). Bear in mind that there are many ways to correct for multiple comparisons, some more well accepted than others (*Eklund et al., 2016*), and therefore the mere presence of some form of correction may not be sufficient.

Further reading
(*Han and Glenn, 2018*; *Noble, 2009*).

## Over-interpreting non-significant results

### The problem

When using frequentist statistics, scientists apply a statistical threshold (normally *alpha*=.05) for adjudicating statistical significance. Much has been written about the arbitrariness of this threshold (*Wasserstein et al., 2019*) and alternatives have been proposed (e.g., *Colquhoun, 2014*; *Lakens et al., 2018*; *Benjamin et al., 2018*). Aside from these issues, which we elaborate on in our final remarks, misinterpreting the results of a statistical test when the outcome is *not* significant is also highly problematic but extremely common. This is because a non-significant *p*-value does not distinguish between the lack of an effect due to the effect being objectively absent (contradictory evidence to the hypothesis) or due to the insensitivity of the data to enable to the researchers to rigorously evaluate the prediction (e.g. due to lack of statistical power, inappropriate experimental design, etc.). In simple words - non-significant effects could literally mean very different things - a true null result, an underpowered genuine effect, or an ambiguous effect (see *Altman and Bland, 1995* for an example). Therefore, if the researchers wish to interpret a non-significant result as supporting evidence *against* the hypothesis, they need to demonstrate that this evidence is meaningful. The *p*-value in itself is insufficient for this purpose. This confound also means that sometimes researchers might ignore a result that did not meet the $p \leq 0.05$ threshold, assuming it is meaningless when in fact it provides sufficient evidence against the hypothesis or at least preliminary evidence that requires further attention.

### How to detect it

Researchers might interpret or describe a non-significant *p*-value as indicating that an effect was not present. This error is very common and should be highlighted as problematic.

*Solutions for researchers*

An important first step is to report effect sizes together with *p*-values in order to provide information about the magnitude of the effect (*Sullivan and Feinn, 2012*), which is also important for any future meta-analyses (*Lakens, 2013*; *Weissgerber et al., 2018*). For example, if a non-significant effect in a study with a large sample size is also very small in magnitude, it is unlikely to be theoretically meaningful whereas one with a moderate effect size could potentially warrant further research (*Fethney, 2010*). When possible, researchers should consider using statistical approaches that are capable of distinguishing between insufficient (or ambiguous) evidence and evidence that supports the null hypothesis (e.g., Bayesian statistics; [*Dienes, 2014*], or equivalence tests [*Lakens, 2017*]). Alternatively, researchers might have already determined a priori whether they have sufficient statistical power to identify the desired effect, or to determine whether the confidence intervals of this prior effect contain the null (*Dienes, 2014*). Otherwise, researchers should not over-interpret non-significant results and only describe them as non-significant.

Further reading
(*Dienes, 2014*).

## Correlation and causation

### The problem

This is perhaps the oldest and most common error made when interpreting statistical results (see, for example, *Schellenberg, 2019*). In science, correlations are often used to explore the relationship between two variables. When two variables are found to be significantly correlated, it is often tempting to assume that one causes the other. This is, however, incorrect. Just because variability of two variables seems to linearly co-occur does not necessarily mean that there is a causal relationship between them, even if such an association is plausible. For example, a significant correlation observed between annual chocolate consumption and number of Nobel laureates for different countries ($r_{(20)}$=.79; $p<0.001$) has led to the (incorrect) suggestion that chocolate intake provides nutritional ground for sprouting Nobel laureates (*Maurage et al., 2013*). Correlation alone cannot be used as an evidence for a cause-effect relationship. Correlated occurrences may reflect direct or reverse causation, but can also be due to an (unknown) common cause, or they may be a result of a simple coincidence.

### How to detect it

Whenever the researcher reports an association between two or more variables that is not due to a manipulation and uses causal language, they are most likely confusing correlation and causation. Researchers should only use causal language when a variable is precisely manipulated and even then, they should be cautious about the role of third variables or confounding factors.

### Solutions for researchers

If possible, the researchers should try to explore the relationship with a third variable to provide further support for their interpretation, e.g. using hierarchical modelling or mediation analysis (but only if they have sufficient power), by testing competing models or by directly manipulating the variable of interest in a randomised controlled trial (*Pearl, 2009*). Otherwise, causal language should be avoided when the evidence is correlational.

Further reading
(*Pearl, 2009*).

## Final remarks

Avoiding these ten inference errors is an important first step in ensuring that results are not grossly misinterpreted. However, a key assumption that underlies this list is that significance testing (as indicated by the *p*-value) is meaningful for scientific inferences. In particular, with the exception of a few items (see 'Absence of an adequate control condition/group' and 'Correlation and causation'), most of the issues we raised, and the solution we offered, are inherently linked to the *p*-value, and the notion that the *p*-value associated with a given statistical test represents its actual error rate. There is currently an ongoing debate about the validity of null-hypothesis significance testing and the use of significance thresholds (*Wasserstein et al., 2019*). We agree that no single *p*-value can reveal the plausibility, presence, truth, or importance of an association or effect. However, banning *p*-values does not necessarily protect researchers from making incorrect inferences about their findings (*Fricker et al., 2019*). When applied responsibly (*Kmetz, 2019*; *Krueger and Heck, 2019*; *Lakens, 2019*), *p*-values can provide a valuable description of the results, which

at present can aid scientific communication (*Calin-Jageman and Cumming, 2019*), at least until a new consensus for interpreting statistical effects is established. We hope that this paper will help authors and reviewers with some of these mainstream issues.

*Further reading*
(*Introduction to the new statistics, 2019*).

## Acknowledgements
We thank the London Plasticity Lab and Devin B Terhune for many helpful discussions while compiling and refining the 'Ten Commandments', and to Chris Baker and Opher Donchin for comments on a draft of this manuscript. We thank our reviewers for thoughtful feedback.

**Tamar R Makin** is a Reviewing Editor for eLife and is in the Institute of Cognitive Neuroscience, University College London, London, United Kingdom; www.plasticity-lab.com
t.makin@ucl.ac.uk
iD https://orcid.org/0000-0002-5816-8979

**Jean-Jacques Orban de Xivry** is in the Movement Control and Neuroplasticity Research Group, Department of Movement Sciences, and the Leuven Brain Institute, KU Leuven, Leuven, Belgium; jjodx.weebly.com
iD https://orcid.org/0000-0002-4603-7939

*Author contributions:* Tamar R Makin, Jean-Jacques Orban de Xivry, Resources, Investigation, Visualization, Writing—original draft, Project administration, Writing—review and editing

*Competing interests:* Tamar R Makin: Reviewing Editor, eLife. The other author declares that no competing interests exist.

## Funding

| Funder | Grant reference number | Author |
|---|---|---|
| Wellcome Trust | 215575/Z/19/Z | Tamar R Makin |

The funders had no role in study design, data collection and interpretation, or the decision to submit the work for publication.

## Additional files

### Data availability
Simulated data and code used to generate the figures in the commentary are available online.

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
