## [Decision Letter]

Thank you for submitting your article "Ten common inferential mistakes to watch out for when writing or reviewing a manuscript" to *eLife* for consideration as a Features Article. Your article has been reviewed by two peer reviewers, who have both agreed to reveal their identity: Nick Parsons; Nick Holmes.

Both reviewers produced very substantial reports, and rather than consolidate them as normally happens at *eLife*, I have combined them so that all the comments on each section of your article are together. This means that the decision letter is very long – and also somewhat critical in a few places – but both referees really engaged with the manuscript

I would like to invite you to submit a revised version of your article that addresses these comments. While the list of comments below is rather long, the reviewers and myself feel that it should be possible to address them within a reasonable time frame.

Also, please note the following:

1) Please remove section 6 (and consider replacing it with a section on 'circular analyses' – as suggested by the referees).

2) Please revise Figure 1 (see comments below).

3) Please remove Figure 3 and Figure 4.

Summary:

Reviewer #1:

Overall, I enjoyed reading this manuscript. It certainly has some merit.

However, at times I found myself profoundly disagreeing with some of the recommendations. I give a list of some of my more significant gripes below. I think this manuscript could be suitable for publication, but there needs to be a substantial amount of additional work to make it make it so.

First, I do not know the background of the authors (so apologies if I offend), but some of the wording and examples and description suggests that they are not themselves experienced applied statisticians. This manuscript would benefit enormously from the input of such a person, simply to reformulate some of the common mistakes and link in with well-known issues that statisticians typically observe when teaching statisticians, advising colleagues and reviewing manuscripts. I accept that it is important to have the tone and voice of the scientist (and not the statistician) in this manuscript, but it is important that the manuscript is such that it is has a much stronger statistical basis, to give it more weight.

Much of the text here is quite wordy and the explanations of the issues often confusing (e.g. issues 3 and 4). I am sure the authors could make these much simpler and easier to understand. The background of the authors is clearly in the neurosciences. This shows with their choice of examples at times (e.g. issue 1). This manuscript would work just as well with more neutral examples that would be understandable to anyone across the range of scientific disciplines. So, I suggest the authors re-write in such a way.

Manuscripts such as this are important and can have significant impact on not just the reporting of science, but also how it is done. So, I hope the authors decide to make appropriate changes to the manuscript in order to make it more acceptable for publication.

Essential revisions:

0) Introduction

Reviewer #1:

– It is worth making the point in the Introduction that many journals undertake in-house statistical reviews and/or send manuscripts out for more detailed statistical review if reviewers of the substantive content have concerns.

1) Absence of a control condition/group

Reviewer #1:

– This manuscript is at times very neuroscience focused. I understand that is the primary interest of the authors, but at times I think it simply distracts from the message and much simpler examples (that would be universal to all scientists) would have worked much better. That is particularly the case with this first common 'mistake'.

Reviewer #2:

– 'inflating the likelihood of observing spurious changes' – But all statistical tests are done using probabilities of false positives, which depend on the variability in the data. What is the evidence that low test-retest reliability leads to increased false-positive outcomes? As this section notes, it is only the *absolute size* of the difference that will be 'observed' – statistics will tell us if this difference is reliable or not, and that is where the (fixed) false-positive rate applies.

– I often come across control groups that are sampled *after* the results of the experimental group are known (e.g., lots of TMS studies). I would add here that control and experimental groups need to be sampled at the same time and with randomised allocation.

– This is not only 'longitudinal', this applies to cross-sectional data too.

2) Interpreting comparisons between two effects without directly comparing them

Reviewer #1:

– Figure 1 – This is an oddly chosen example. Clearly the variance in group B is much greater than the variance in group A. This explains (in part at least) why the test of the group differences is not significant. But, surely pooling here is problematic: we are assuming, for the methods suggested, that the variance is the same in each group whereas, to most, it looks like it is very different. The authors need to choose a better example to illustrate common mistake 2, and modify Figure 1 appropriately.

Reviewer #2:

– This problem applies also not just to 'difference scores' but any effect (e.g., a slope, curve-fit etc., not just 'differences'). I suggest the authors make it more general here (as they do in 'how to detect'), then give the specific and useful example of the simple difference of two differences. It may also be worth noting here that this is often the 'interaction' term in the analysis.

– 'differential statistical significance' – I would say something like 'different binary outcomes when applying a statistical threshold'.

3) Inflating degrees of freedom by violating independence of measures

Reviewer #1:

– This is a very complicated explanation for what most statisticians would describe in a very different way. It is needlessly complicated. This is what most statisticians would describe as the 'unit of analysis' issue – much described in the literature previously see e.g. Parsons, Teare and Sitch (2018, *eLife*).

Some poor practice is described here, where for example multiple measurements on the same subject are made as a means of ultimately comparing subjects. If a study aims to understand the effect of an intervention on subjects, then that is the 'unit of the analysis' and in order to draw inferences the replication must be at the level of the subject (the unit of analysis), not within subject (within unit). Multiple measurements on subjects improve the precision of estimation of the subject mean (for instance) but tell us nothing about the variability between subjects.

This is often observed as an artificial inflation of the degrees of freedom, pooling between strata in the analysis, but ultimately the problem is the lack of clear identification of the purpose of the analysis and the appropriate unit to use to assess variation that is used to quantify intervention effects. Personally, I don't think bringing correlation into the discussion helps a great deal. All we really need to be aware of is that measurements within (for instance) a subject are likely to be correlated, whereas by definition data from subjects are uncorrelated.

– Mixed-effects analysis – This is the canonical analysis that most statisticians would recommend. When we do this, we naturally estimate the appropriate within subject (cluster) correlations. This methodology should be much more widely used in many areas of science; only in medicine, where studies report patient data, and psychology is it generally recognised as being important. Although its roots go back to the genesis of statistics in agricultural science, where fields were divided into blocks, plots and nested sub-plots and plants etc.

– A priori statistical power analyses are always a good idea, but I really don't think it adds much to the discussion here.

Reviewer #2:

– This is true in some statistical procedures (e.g., where you model a single parameter at a single level for each participant), but not all. For example, I believe linear mixed models (e.g., in R), will have many *df*s larger than N-x, yet these remain valid. I remember seeing large *df*s in (e.g.) Brain Voyager FMRI outputs. I do not understand these multi-level linear mixed models, but I have questioned and been corrected on this *df* point by statisticians using R (e.g., see the *df* in Meteyard and Holmes, 2018; I didn't do the analysis). The recommendation in 'how to detect' should be clarified and/or corrected as necessary.

– 'average the resulting r values (don't forget to normalise the distribution first!)' – Perhaps give specific advice here: e.g., use Fisher's r-to-Z transformation, Z=0.5log[(1+r)/(1-r)]

– 'random factor' – 'random 'effect' would be better

4) Spurious correlations

Reviewer #1:

– Figure 2A, B, C – Surely the issue here is that 'outliers' have a big impact (leverage) on many statistics; means, variances, covariances, regression analyses, ANOVA and yes, correlations. But this is not an issue only of concern when estimating correlations. Surely the issue here is to present data visually and consider the meaning (validity) of any data points that are a long way from the rest of the distribution. I don't like the (implicit) argument here that the Pearson correlations are in some sense 'wrong'. It depends on whether the model is correct (straight line) and whether the assumptions of approximately normality are correct. It is perfectly plausible to believe Figure 2C is correct, and that only one data point was available at X=5, but there is good reason to believe that data are normally distributed. The point I would make here is the importance of error measurements when reporting. The confidence intervals of the Pearson correlation would help us enormously here. Point estimates of correlations alone are not that useful, unless the data are shown visually.

– The other thing I would take issue with here is the implication that Spearman's rank correlation makes more sense in settings Figure 2B and Figure 2C. In general, decisions about whether to assume normality are better made for principled reasons rather than for empirical reasons. Using a non-parametric correlation coefficient would make little sense to me here – they are generally very inefficient, as we convert to ranks first, which is the reason the value does not change from Figure 2B to Figure 2C. If data were reasonably tightly distributed symmetrically about the mean, other than one value which was a big distance away, my first recommendation would be to examine the credence of the extreme data point, not proceed to a non-parametric correlation.

Reviewer #2:

– 'Yet, the use of parametric correlations, such as Pearson's r, requires that both variables are normally distributed.'

No, it doesn't! All parametric linear models (as far as I understand) require that the *error* is normally distributed. In the case of a single-sample t-test against a single mean, this is identical to the requirement that the variables themselves are normally distributed. But, for everything else, it is the *differences* or *error* or *residuals* after the model is fit which must be normally distributed, not the raw data. The authors repeat in their tutorial what I understand to be a very common mis-interpretation, and it would be good for them to make absolutely certain that what they say here is correct, to avoid perpetuating these errors.

Here is a tutorial from the R team: https://rcompanion.org/handbook/I_01.html, specifically: "In particular, the tests discussed in this section assume that the distribution of the data are conditionally normal in distribution. That is, the data are normally distributed once the effects of the variables in the model are taken into account. Practically speaking, this means that the residuals from the analysis should be normally distributed. This will usually be assessed with a histogram of residuals, a density plot as shown below, or with a quantile-quantile plot… Be careful not to get confused about this assumption. You may see discussion about how "data" should be normally distributed for parametric tests. This is usually wrong-headed."

The authors are correct here that (genuine) outliers can lead to spurious correlations, but the remedy for this is, as they state: (a) plot the data, (b) run some robustness-checks, and to report all the results with their standard errors and with due caution.

One real problem is how do we identify 'genuine' outliers? Perhaps a large sample size is one remedy, so that we have a better coverage of the population? Yet, there will still be cases when clear 'outliers' are genuine observations which obey the law that you are trying to discover. For example, measuring mass vs. body length across the animal kingdom: there will be an awful lot of small animals down the bottom of the scale (e.g., insects), some in the middle (e.g., birds and most mammals), and fewer still at the extremes (e.g., whales or elephants). I would bet that the blue whale follows the same statistical law of mass vs. length as the gnat (with variance away from this model due to shape). A 'spurious' correlation might arise from incomplete sampling of the problem space – if we only sampled insects and whales, we might draw the wrong conclusion and call the correlation between mass and length 'spurious'. Or perhaps the data need log-transforming first?

My laboured point here is: if you don't have any *independent* reason to exclude a particular datapoint (e.g., the participant didn't do the task properly, wasn't wearing their glasses, is not healthy, is not typical; the elephant was stretching its legs), then I think it is dangerous to conclude that the correlation is spurious just because of one 'outlier'. Rather, authors need to present the data, check their assumptions, and speculate that sampling bias or experimenter error has led to this 'outlier'. In general, the authors are correct here: limited sampling of the intended population may make such outliers more likely, and correlations may then be more problematic. But this is all relative, and error can occur in both directions (Type I, Type II). The only solution is to be very careful, both in including and excluding data.

– 'when the two variables are not independent' – Repeated measures designs will often have highly-correlated scores between different conditions or time-points in the same participants. Again, is the real issue here 'independent error' (residuals) not 'independent data'?

– This is the 'regression towards the mean' error that I discussed in Holmes, (2007, 2009), yet this topic is only an "Honorable mention" here! I would suggest all these 'circular analyses' and 'double-dips' (i.e., both are experimenter-created dependencies in the data) could be in their own section (after dealing with the below comments, in which I suggest removing point 6 entirely).

– Figure 2A – I would bet that the red point is not an outlier here, as claimed in the legend.

– 'they can run some basic simulations' – I agree 100% (Holmes 2007, 2009), but this sentence will have, in my view, about 95% of your target audience hiding under the bedcovers in fear of programming. How does someone who is not sufficiently well-trained to spot these problems in the first place go about 'running some simulations'? Can the authors point to an online tool or tutorial that helps?

5) Use of underpowered groups

Reviewer #1:

– Experiments with small samples sizes are quite often small for very good reasons, not always but often. We should not recommend that scientists don't do small experiments – sometimes there is no option – but we should tell them *not* to report inferential statistics. Particularly if the study does not have an a priori power calculation. Not sure that Figure 3 adds much here.

Reviewer #2:

– 'In frequentist statistics in which a significance threshold of α=.05 is used, 5% of all statistical tests will yield a significant result even in the absence of an actual effect (false positives; Type I error)' – I think the authors need to clarify this a bit more, to, e.g.: "Assuming that the null is true, then randomly- and independently-sampled data from a normal distribution with a mean of zero will yield a sample that, when tested against a mean of zero, has a p-value below or equal to. 05 approximately 5% of the time." The word 'even' in their claim here is unhelpful – the stats explicitly assume that the null *is* true (it is *never* actually true!)

– 'Given that these two variables are random, there should be no significant correlations' – See previous point. There will be 5% 'significant' correlations.

– 'falsely significant' – I don't like this phrase. It seems contradictory. I know what they mean (something like: 'using the standard α criterion, most researchers would conclude that there is a positive correlation in the population when in fact there isn't').

– 'the experiment is underpowered' – But there is no effect in the simulated population. There can therefore be no sample size sufficient to find this effect. This cannot, therefore, be 'underpowered'. Revise.

– '<' should this, in fact, be '≤'? Same throughout the manuscript.

– 'Designs with a small sample size are also more susceptible to Type II errors' – Why? Type II error is a non-linear function of the sample size and the real effect size. Knowing the Type II error requires that you know the population distribution, which is almost never the case (and not required) in the kinds of parametric null-hypothesis tests that the authors are discussing here.

– 'based on a limited number of participants' – I would remove this as I don't think this it is justifiable. I think all effects (especially surprising ones) from a single experiment should be taken with the same degree of caution, regardless of their sample size (who sets the criterion in any case?). The statistics deal with the problem of sample size. Statistics can be biased as a function of sample size, of course, and some come with corrections (e.g., Hedges G instead of Cohen's d) but if you expect a large effect (e.g., removing striate cortex will impair vision), then I see nothing wrong with doing the absolute minimum of testing on your subjects to establish that effect. It would be unethical to remove 30 monkeys' visual cortices when 2 are sufficient to test the hypothesis.

– 'that were not replicated' – Yes, we should be *especially* skeptical if a second well-powered experiment failed to replicate the first, but I would only be *normally* skeptical of a single-experiment finding, regardless of its sample size. I think the authors' main point is that small studies will achieve significance only with large effects. True. But some large effects are real, so given a single particular result, how do you know? We *must* be allowed to search for large statistical effects. Collecting converging and independent evidences should be sought in all investigations, not just in those researchers looking for large effects: Smith and Little (2018).

6) Using parametric statistics with small sample sizes

Reviewer #1:

– Sorry but I cannot agree with much of this section. See my previous comments about normality assumptions. Tests of normality are really *not* very useful in most circumstances. In small samples, I agree with authors, they are useless. However, in large samples they will always reject with probability of very near one.

– In general, decisions about normality and whether to use parametric or non-parametric methods should be based mainly on scientific principle. For instance, if I collect data on the heights of 10 people, I report a median and IQR, but if I collect data on 50 people a mean and SD? No, that is clearly wrong. I believe that heights are approximately distributed, based on the way it is measured, my own experience and the experience of others (irrespective of what a test of normality tells me!), so I should summarise data on that basis in the appropriate way by a mean and SD.

– This sort of mechanical/automated approach to the implementation of statistical methods is strongly discouraged by the majority of statisticians. This is analogous to the widespread adherence to the (mis-) interpretation of p-values that has been so widely criticized by among others the American Statistical Association. In its guidance for the use of statistical tests for such decisions and the role of p-values it makes clear that "Scientific conclusions and business or policy decisions should *not* be based *only* on whether a p-value passes a specific threshold" and "No single index [the p-value] should substitute for scientific reasoning".

– Scientific reasoning and precedent should be used to make decisions about how to appropriately analyse data, not arbitrary ad hoc (data dependent) statistical tests.

Reviewer #2:

– Section 6 is incorrect and should be removed. In part, due to the same problem as point #4 (normal distribution of error), but, what the authors seem to be arguing for (unwittingly) is an *abandonment* of parametric tests, not for their use only with 'large' samples, as explained below:

Parametric statistics were *designed for small samples*. If in any doubt about this, please check out Fisher (1925), available in full here: https://psychclassics.yorku.ca/Fisher/Methods/index.htm

In parametric statistics, from a sample of data we extract parameters, for example the mean and the SD (e.g., of the difference between means of two conditions). In our statistical model we compare those parameters to a normal distribution in order to make probabilistic inferences using the theoretical distribution.

When sample size is small (say <30), we do not refer our parameters directly to the normal distribution. Rather, we refer them to the t, F, Chi-Square, Poisson, Binomial, or any other appropriate statistical distribution. These distributions were originally created by sampling small numbers of datapoints, and seeing how they behaved. These distributions are all skewed in interesting ways, but as N increases, they tend to approximate the normal distribution.

The (in my view) often-mistaken 'rule of thumb' that you need at least 30 participants to do parametric statistics is wrong. It is in my understanding, *opposite* this. As N increases, the t-, F, Binomial, Chi-square, and Poisson distributions converge closer and closer to the normal distribution. So, when N=30, rather than using the t-test, you can just use the Z-test (i.e., essentially ignoring sample size). The critical t-value for 1 degree of freedom (N=2) at α=.05 is 6.31 (i.e., 6.31 standard errors of your sample mean difference away from zero). As N increases to infinity, the critical value converges to 1.645. At N=30, the critical t-value is 1.7, which is arguably close-enough to the population Z-score (1.645) that the t-distribution can be abandoned (i.e., only sample size is relevant for calculating the SE, *df* is not needed) and that the Z-distribution can be used instead. Small samples are already 'punished' via the *df*, by requiring much larger effect sizes to pass arbitrary statistical thresholds.

My understanding is that most statistical tests are designed for small samples. Non-parametric tests are for non-interval and non-ratio data (categorical, ordinal), or for interval/ratio data with populations for which no reasonable assumptions can be made (e.g., with large inexplicable outliers). Bootstrapping or other non-parametric statistical methods can be useful to check whether small samples are indeed sufficiently normal to use parametric tests (e.g., from Makin et al., 2009, where Ns=6-11: "In every case, this bootstrapping procedure supported the inferences derived from the t tests, so we report only the standard parametric tests in this manuscript.").

As the authors correctly note, small samples come with biases (e.g., effect sizes are larger for significant effects), but this does *not* invalidate the use of parametric tests. The authors cite Kar and Ramalingam (2013) in support of their claim, yet from that paper's conclusion: "Hence, there is no such thing as a magic number when it comes to sample size calculations and arbitrary numbers such as 30 must not be considered as adequate."

– 'though it is well agreed that you shouldn't use a parametric test with N<10 (Fagerland, 2012)' – The authors present no evidence for this 'well agreed' rule. I disagree with it, for example, as I believe would the statisticians who invented parametric tests. The article cited by Fagerlund, (2012) was specifically looking at *skewed* distributions (gamma and log-normal) of the underlying population parameters, and made the same common mistake about normal distribution of the data (should be: error/residuals). Yes, with data producing skewed error distributions, transformation or non-parametric tests are required, *or* a larger sample, and the central limit theorem can be relied upon. It is absolutely fine to use parametric tests under reasonable assumptions and with reasonable caution with N as low as 2 (Fisher, 1925).

– 'Nonparametric tests… are less sensitive to outliers' – Bootstrapping is a kind of non-parametric test. Such tests may *highlight* outliers by revealing a multi-modal distribution of summary statistics, but they are just as 'sensitive' to outliers. Inspecting the distributions and checking assumptions is the correct approach.

7) Flexibility of analysis: p-hacking

Reviewer #1:

– The authors advice on how to detect p-hacking is well-meaning but naïve.

In truth, only by pre-registering and providing detailed analysis plans, such as we do in clinical trials, can we ever hope to stop p-hacking. Almost impossible for a reviewer to make much assessment of this, unless they have a study protocol available against which to assess the reporting adherence.

8) Failing to correct for multiple comparisons

Reviewer #1:

– This is a tricky one. The truth is that there is no great consensus amongst statisticians as to the best correction method to use. It is very application area dependent and there are many who would simply disagree on principle that correcting for multiple testing makes any sense (e.g. Rothman, 1990, Epidemiology).

I would draw a distinction between exploratory and confirmatory analyses, and make differing recommendations dependent on the aims of the study. We may be more or less worried about false negatives and false positives in these settings.

This topic is complex and probably beyond the knowledge of most (non-expert) reviewers and beyond the scope of this article.

Reviewer #2:

– 'it is simply unacceptable for the researchers to interpret results that have not survived correction for multiple comparisons' – Even if hypothesised? Perhaps 'exploratory' needs to be added here. I disagree that if an effect 'could' be tested using different comparisons, then corrections for multiple comparisons are required. A reviewer could just say: “well, you *could* have done this on all the individual blocks of data rather than the subject averages, so you need to correct…” This would be r-hacking (reviewer hacking), and needs to be discouraged. Explicit, limited, pre-registered hypothesis-testing should be encouraged. Exploratory testing is fine, but should be acknowledged and α corrected.

9) Over-interpreting non-significant results

Reviewer #1:

– This is a common error – so common in fact that it is hard to believe anything we suggest will make much difference! The suggestions of the authors are reasonable, but a bit wishy-washy.

Reviewer #2:

– 'non-significant effects could literally mean anything' – So could significant effects. All the problems listed apply equally to significant effects: true positives, over-powered small effects (e.g., much smaller than the meaningful effect that a theory predicts), or ambiguous effects. There is nothing special about the α value, as the authors note.

– 'Otherwise, researchers should not over-interpret non-significant results and only describe them as non-significant.' – So, p=.049 is "significant" and can be interpreted, and p=.051 is "non-significant" and should 'not be over-interpreted'. I think we can do better than this. What rules of thumb do the authors offer to get around this linguistic threshold? My view would be that if there is any doubt in a particular result, then plot the data, check assumptions, run simulations, replicate the experiment with increased power, seek converging evidence, do a systematic review and meta-analysis, present the work at conferences, ask reviewers… Being told to stick rigorously to the 'significant/non-significant' dichotomy is not going to improve the readers' statistical inferences.

– Figure 4 does not show a 'correlation' but two time series data; the right y-axis looks like negative numbers because of the axis ticks; the blue dataset has auto-correlation (mostly the same people eating margarine across different years) but the red does not (mostly different people getting divorced). Since the authors did not create this figure, I suggest removing it from their tutorial. I also suggest they cite primary research data, rather than secondary websites (particularly when that website labels a correlation of r[without degrees of freedom]=.9926 as a '99.26%' correlation. It's not).

10) Correlation and causation

Reviewer #1:

– My sense is that scientists generally have a good understanding of this issue. Not really a statistical issue per se, more about using cautious language when reporting.

Reviewer #2:

– 'Impossibly high correlations' – Replace with 'effect sizes'?

Honourable mentions

Reviewer #1:

– I really don't think section adds much, just a list of terms with little or no more explanation. I would advise deleting.

Conclusions

Reviewer #1:

– Interesting as it is, I don't see why we need a discussion of NHST and p-values as the conclusion. Seems a bit off-topic. A summary of main issues and overlap of the common mistakes and importance would be much more useful. And some recognition of the importance of talking to your statistical colleagues. Most of the issues discussed here are very common issues that all (any) statisticians will be well placed to help with. Whether at study development, writing or reviewing stages of research.

[Editors' note: further revisions were requested prior to acceptance, as described below.]

Thank you for submitting the revised version of "Ten common inferential mistakes to watch out for when writing or reviewing a manuscript" for consideration by *eLife*. This version has been seen by the two reviewers who reviewed the original version (Nick Parsons and Nick Holmes), and their comments are below. It should be straightforward to address these comments, so I would like to invite you to submit a second revised version that addresses these comments.

*Reviewer #1:*

The authors have clearly worked tremendously hard to make changes to the manuscript. It is now quite difficult to read, given all the additions and deletions, so will need a good proof-read to make sure it still makes good sense and scans properly. I make a number of responses to these changes:

Summary:

I still think it is a strange argument to state initially that this paper is motivated by "…ineffective experimental design, inappropriate statistical analysis, and/or flawed reasoning, appearing in published neuroscience papers…", and then to say a little later that all the issues highlighted are "…applicable across a range of scientific disciplines that use statistics to assess findings…". The latter is true – the issues highlighted are very familiar to most applied statisticians who work in science. If there are particular issues that relate to neuroscience – which I imagine there may well be – then that does not come across at all in this manuscript. For me, I think this manuscript would have worked much better if 'neuroscience' had been added to the title and examples and issues specific to neuroscience had been used throughout. Overall, I do not think that this manuscript really delivers on its aim of responding to and talking directly to neuroscience readers. The issues picked-up are the usual suspects; the kind of issues that statistical reviewers and applied statisticians are very familiar with. Nothing wrong with that, per se, but maybe a missed opportunity to do something more impactful. For instance, a survey of the published literature, and description of common reporting and analysis errors would have been an excellent way of motivating this manuscript. Not seriously suggesting that this be done now. But this has proved to be a highly effective way of making real changes to the research culture and the way research is done and reported in other disciplines.

0) Introduction

OK. This is clearly a matter of perspective. I get asked to review papers, with a specific request asking me to look at statistical issues, quite frequently. These often come after requests from non-statistical reviewers who suggest the editor gets a statistician to look at the manuscript.

2) Interpreting comparisons between two effects without directly comparing them

Not really convinced that this is type of "…erroneous inference.…" is "…very common…" in published papers. Almost the first thing we teach in statistics is how to compare group A to group B, using an appropriate statistical test. The error that the authors highlight here feels much more complicated than this; to compare the mean response in two groups, would I really test each against the null hypothesis that the mean is 0, and then conclude if I reject for one group, then I can infer that this group is 'statistically significantly' different to the other group? If this is done, then it is done to deliberately (maliciously) mislead the reader.

4) Spurious correlations

The point I was trying to make about adding confidence intervals seems to have been misunderstood. I would make the general point that, if possible (which it generally is), *all* point estimates of quantities should be presented with errors; e.g. CIs, range, standard error, bootstrapped intervals etc. The point being that if this had been done for the correlations, then the effect of the outlier on inferences would be much more obvious than it is in a plot that simply presents a point estimate of a correlation as a straight line. The statement that the points in red are clear 'outliers' – presumably because they are a long way from the fitted line – would be much less sustainable as an argument if the line were actually a region of plausible values, given the observed data.

I am absolutely *not* suggesting that data points should be discarded based simply on post-hoc visualisation of the data. The critical point is that scientists need to always question their own data – and not just at the end of a study when all the data have been collected and they no longer remember why one value is far away from all the others. It is perfectly acceptable to discard values, if there is good reason to believe that something has gone wrong or been recorded incorrectly. This happens all the time – e.g. numbers accidently recorded in the wrong units, calibration not done, days and months confused in dates – there is an almost infinite number of ways that data can be 'wrong'. It is perfectly acceptable to change values such as this – I would advise that the raw data remain unchanged and edits are made in a revised dataset, with changes documented and agreed by all in the study, and made available for others to inspect. If we do this, then what do we make of the data-points the authors highlight as 'outliers'? They are either true errors (things went wrong, but we can't find a reason) or then true data-points. In my view, neither case is a good reason to suggest using robust-correlations, if the rest of the data look reasonably normally distributed. Necessary if you believe the latter is true to modify the model being used, or in the former maybe restrict inferences (model fitting) to the region where you have good data, and not include the extreme value(s).

It is not correct to believe that for instance using a non-parametric method is a solution (good alternative) to a parametric method, in the setting described here. As I stated in my original comment, because values are converted to ranks you simply move the extreme value in Figure 2C closer to the other values. So really what you are doing is saying that you do *not* believe the data as recorded are correct in the sense that they can be treated in the way that is implied by the plots; i.e. that they are continuous measures, where the distance between them has some sense of measuring the 'closeness' (e.g. Euclidean distance). So if you use a non-parametric correlation you are really saying that you do not believe the value is 'correct'. Ultimately this goes right back to the basics of designing an experiment and writing a statistical analysis plan (SAP) at the start of a study before data collection begins. In that case you need to make clear what your beliefs are about the metric properties of each study outcome – you cannot simply choose to use a non-parametric analysis after the data are collected because that makes life easier. Sorry to labour the point, but this is important, as it goes to the heart of many of the statistical problems in reporting of science.

*Reviewer #2:*

The commentary/tutorial is more nuanced, fairer, more correct, and thus more publishable. I think it will make a nice addition to a large and very long history of statistical advice to researchers.

---

## [Author Response]

Summary:Reviewer #1:Overall, I enjoyed reading this manuscript. It certainly has some merit.However, at times I found myself profoundly disagreeing with some of the recommendations. I give a list of some of my more significant gripes below. I think this manuscript could be suitable for publication, but there needs to be a substantial amount of additional work to make it make it so.First, I do not know the background of the authors (so apologies if I offend), but some of the wording and examples and description suggests that they are not themselves experienced applied statisticians. This manuscript would benefit enormously from the input of such a person, simply to reformulate some of the common mistakes and link in with well-known issues that statisticians typically observe when teaching statisticians, advising colleagues and reviewing manuscripts. I accept that it is important to have the tone and voice of the scientist (and not the statistician) in this manuscript, but it is important that the manuscript is such that it is has a much stronger statistical basis, to give it more weight.Much of the text here is quite wordy and the explanations of the issues often confusing (e.g. issues 3 and 4). I am sure the authors could make these much simpler and easier to understand. The background of the authors is clearly in the neurosciences. This shows with their choice of examples at times (e.g. issue 1). This manuscript would work just as well with more neutral examples that would be understandable to anyone across the range of scientific disciplines. So, I suggest the authors re-write in such a way.Manuscripts such as this are important and can have significant impact on not just the reporting of science, but also how it is done. So, I hope the authors decide to make appropriate changes to the manuscript in order to make it more acceptable for publication.

We thank the reviewer for this comment, as it highlights the need for us to be more explicit from the onset as to our intentions and target audience with regards to our commentary. This commentary is written by neuroscientists to their neuroscientist peers and their trainees (please note that we envisaged it to be featured under the ‘neuroscience’ section). We now clarify that we are neuroscientists rather than statisticians. In fact, the issues we highlight require minimal statistical training, which is why we believe this comment can make a real impact. Considering our training and perspective, we do not feel like we are well placed to write a general list for the issues that are shared across fields of science, we therefore believe that targeting the neurosciences will be the most appropriate and effective approach. We therefore wish to keep the examples and general discussion accessible and relevant for our target audience, though we have taken this comment on board, and when possible, we simplified the examples or reduced some of the real-life details.

Abstract: “Inspired by broader efforts to enhance the rigor of the different phases of scientific research, here we compile a list of some of the most common inference errors due to ineffective experimental design, inappropriate statistical analysis, and/or flawed reasoning, appearing in published neuroscience papers.”

Introduction: “Here we highlight some of the most common and pertinent errors in scientific interpretation that should be addressed during the peer review process. […] We hope that this list will help sharpen understanding of why these issues are problematic, how to detect them in a manuscript and how to address them in the review process.”

Essential revisions:0) IntroductionReviewer #1:– It is worth making the point in the Introduction that many journals undertake in-house statistical reviews and/or send manuscripts out for more detailed statistical review if reviewers of the substantive content have concerns.

Considering the focus on general neuroscience, in our experience this is quite uncommon for general research papers (this might be more commonly practices in some clinical journals?). This might be the reason why the neuroscience community is plagued with these inference errors.

1) Absence of a control condition/groupReviewer #1:– This manuscript is at times very neuroscience focused. I understand that is the primary interest of the authors, but at times I think it simply distracts from the message and much simpler examples (that would be universal to all scientists) would have worked much better. That is particularly the case with this first common 'mistake'.

Agreed. We adapted the text to more broadly suit various sub-disciplines of the neurosciences:

“For instance, when examining the effect of training, it is common to probe changes in behaviour or a physiological measure. Yet, changes in outcome measures can arise due to other elements of the study that do not directly relate to the manipulation (e.g. training) per se. Repeating the same task in the absence of an intervention might induce a change in the outcomes between pre- and post-intervention measurements, e.g. due to the participant or the experimenter merely becoming accustomed to the experimental setting, or due to other changes relating to the passage of time.”

Reviewer #2:– 'inflating the likelihood of observing spurious changes' – But all statistical tests are done using probabilities of false positives, which depend on the variability in the data. What is the evidence that low test-retest reliability leads to increased false-positive outcomes? As this section notes, it is only the absolute size of the difference that will be 'observed' – statistics will tell us if this difference is reliable or not, and that is where the (fixed) false-positive rate applies.

Agreed. We have removed the following sentence from the manuscript: “If the test-retest reliability is low, then natural fluctuations of the variable over time will be large, thereby inflating the likelihood of observing spurious changes over time.”

– I often come across control groups that are sampled after the results of the experimental group are known (e.g., lots of TMS studies). I would add here that control and experimental groups need to be sampled at the same time and with randomised allocation.

Agreed, this is indeed very relevant, we have now mentioned this malpractice in the text as follows:

“It is also important that the control and experimental groups are sampled at the same time and with randomised allocation, to minimise any biases.”

– This is not only 'longitudinal', this applies to cross-sectional data too.

Agreed. We have taken ‘longitudinal’ off the following sentence: “Conclusions are drawn on the basis of longitudinal data of a single group, with no adequate control conditions.”

2) Interpreting comparisons between two effects without directly comparing themReviewer #1:– Figure 1 – This is an oddly chosen example. Clearly the variance in group B is much greater than the variance in group A. This explains (in part at least) why the test of the group differences is not significant. But, surely pooling here is problematic: we are assuming, for the methods suggested, that the variance is the same in each group whereas, to most, it looks like it is very different.The authors need to choose a better example to illustrate common mistake 2, and modify Figure 1 appropriately.

Point taken. We have revised this figure to convey two very common examples. We deliberately picked up extreme examples (which are actually based on a real publication), so that our key point can be immediately appreciated.

Reviewer #2:– This applies also not just to 'difference scores' but any effect (e.g., a slope, curve-fit etc., not just 'differences'). I suggest the authors make it more general here (as they do in 'how to detect'), then give the specific and useful example of the simple difference of two differences. It may also be worth noting here that this is often the 'interaction' term in the analysis.

Agreed. This sentence has been modified accordingly, and the figure exemplifying this issue has been changed to include two different cases.

“Researchers often base their conclusions regarding the impact of an intervention (such as a pre- vs. post-intervention difference, correlation between two variables, etc.) by noting that the intervention yields a significant effect whereas the corresponding effect in the control condition or group is not significant (Nieuwenhuis et al., 2011). Based on this evidence, researchers will sometimes suggest that the effect is larger in the experimental than the control condition. This type of erroneous inference is very common but incorrect.”

– 'differential statistical significance' – I would say something like: 'different binary outcomes when applying a statistical threshold'.

Agreed. This has now been rephrased as follows:

“For instance, as illustrated in Figure 1A, two variables X and Y, each measured in two different groups of 20 participants could have a very similar correlation (group A: R=0.47; group B: R = 0.41) but different outcomes in terms of statistical significance: the two variables for group A meet the statistical threshold *p*≤0.05 for achieving significance but not for group B. […] However, it does not mean that the effect of the intervention is different between the two groups; indeed in this case, the two groups do not significantly differ.”

3) Inflating degrees of freedom by violating independence of measuresReviewer #1:– This is a very complicated explanation for what most statisticians would describe in a very different way. It is needlessly complicated. This is what most statisticians would describe as the 'unit of analysis' issue – much described in the literature previously see e.g. Parsons, Teare and Sitch (2018, eLife).Some poor practice is described here, where for example multiple measurements on the same subject are made as a means of ultimately comparing subjects. If a study aims to understand the effect of an intervention on subjects, then that is the 'unit of the analysis' and in order to draw inferences the replication must be at the level of the subject (the unit of analysis), not within subject (within unit). Multiple measurements on subjects improve the precision of estimation of the subject mean (for instance) but tell us nothing about the variability between subjects.This is often observed as an artificial inflation of the degrees of freedom, pooling between strata in the analysis, but ultimately the problem is the lack of clear identification of the purpose of the analysis and the appropriate unit to use to assess variation that is used to quantify intervention effects. Personally, I don't think bringing correlation into the discussion helps a great deal. All we really need to be aware of is that measurements within (for instance) a subject are likely to be correlated, whereas by definition data from subjects are uncorrelated.

We thank the reviewer for their very helpful account of the problem. We now elude to the useful reference offered by the reviewer, and have re-written the section to reflect this as a ‘unit of analysis’ issue. The correlation example is a true example (from an *eLife* publication, as a matter of fact!). This is an unbelievably common issue in our field, and we felt like an example is needed to give people a more intuitive sense of the issue (and to why people might fall prey to it – it improves their statistical outcomes).

– Mixed-effects analysis – This is the canonical analysis that most statisticians would recommend. When we do this, we naturally estimate the appropriate within subject (cluster) correlations. This methodology should be much more widely used in many areas of science; only in medicine, where studies report patient data, and psychology is it generally recognised as being important. Although its roots go back to the genesis of statistics in agricultural science, where fields were divided into blocks, plots and nested sub-plots and plants etc.

Agreed. We have highlighted this solution in our original manuscript, and expanded on this in the revisions. While this is probably the best statistical solution to this problem, it is also one that requires some advanced statistical understanding to implement and therefore should be practiced with caution. As such, it is probably not readily available to our target readership.

“Perhaps the best available solution to this issue is using a mixed-effects linear model, where researchers can define the variability within subjects as a fixed effect, and the between-subject variability as a random effect. This increasingly popular approach (Boisgontier and Cheval, 2016) allows one to put all the data in the model without violating the assumption of independence. However, it can be easily misused (Matuschek et al., 2017) and requires advanced statistical understanding, and as such should be applied and interpreted with some caution.”

– A priori statistical power analyses are always a good idea, but I really don't think it adds much to the discussion here.

Agreed. We have removed this suggestion from this section.

Reviewer #2:– This is true in some statistical procedures (e.g., where you model a single parameter at a single level for each participant), but not all. For example, I believe linear mixed models (e.g., in R), will have many dfs larger than N-x, yet these remain valid. I remember seeing large dfs in (e.g.) Brain Voyager FMRI outputs. I do not understand these multi-level linear mixed models, but I have questioned and been corrected on this df point by statisticians using R (e.g., see the df in Meteyard and Holmes, 2018; I didn't do the analysis). The recommendation in 'how to detect' should be clarified and/or corrected as necessary.

We have now clarified that in this specific example the researchers are carrying out a simple regression analysis. We also reframed this issue as per reviewer #1’s suggestion around ‘units of analysis’, thus minimising our discussion of *df*.

– 'average the resulting r values (don't forget to normalise the distribution first!)' – Perhaps give specific advice here: e.g., use Fisher's r-to-Z transformation, Z=0.5log[(1+r)/(1-r)].

We rephrased this sentence, but have not included the formula, as it seems too detailed for the purpose of this section.

– 'random factor' – 'random 'effect' would be better.

Changed.

4) Spurious correlationsReviewer #1:– Figure 2A, B, C – Surely the issue here is that 'outliers' have a big impact (leverage) on many statistics; means, variances, covariances, regression analyses, ANOVA and yes, correlations. But this is not an issue only of concern when estimating correlations. Surely the issue here is to present data visually and consider the meaning (validity) of any data points that are a long way from the rest of the distribution. I don't like the (implicit) argument here that the Pearson correlations are in some sense 'wrong'. It depends on whether the model is correct (straight line) and whether the assumptions of approximately normality are correct. It is perfectly plausible to believe Figure 2C is correct, and that only one data point was available at X=5, but there is good reason to believe that data are normally distributed. The point I would make here is the importance of error measurements when reporting. The confidence intervals of the Pearson correlation would help us enormously here. Point estimates of correlations alone are not that useful, unless the data are shown visually.

Agreed, and in this section, we are highlighting the advantages of robust correlations, which take the variance of a given distribution into account. But to the average neuroscientist, the CI by itself will not mean much (in fact, most journals outside the psychological ones will not require/encourage the authors to report the descriptive statistics). Moreover, as the reviewer indicates, the CI of the correlation (which we have now added to the figure) will not allow you to determine whether a given correlation is spurious or not – this is a deeper issue, as we elaborate on below.

“Spurious correlations most commonly arise if one or several outliers are present for one of the two variables. As illustrated in the top row of Figure 2, a single value away from the rest of the distribution can inflate the correlation coefficient. Spurious correlations can also arise from clusters, e.g. if the data from two groups are pooled together when the two groups differ in those two variables (as illustrated in the bottom row of Figure 2).”

– The other thing I would take issue with here is the implication that Spearman's rank correlation makes more sense in settings Figure 2B and Figure 2C. In general, decisions about whether to assume normality are better made for principled reasons rather than for empirical reasons. Using a non-parametric correlation coefficient would make little sense to me here – they are generally very inefficient, as we convert to ranks first, which is the reason the value does not change from Figure 2B to Figure 2C. If data were reasonably tightly distributed symmetrically about the mean, other than one value which was a big distance away, my first recommendation would be to examine the credence of the extreme data point, not proceed to a non-parametric correlation.

We strongly agree that any decisions about statistical analysis and outlier removal should be decided a priori and for principled reasons (see our section on p-hacking). For these reasons, we don’t think data points should be discarded based on post-hoc visualisation of the data, we believe this will be a point of consensus between us and the reviewers. So, the problem at hand is – how to deal with a situation where the results seem to be driven by an outlier/cluster, without opening Pandora’s box of p-hacking? Exasperating this problem is the fact that in many sub-filed of neuroscience the sample sizes are very limited, making it difficult to determine if the data violates the assumptions of parametric statistics, including “true” outliers identification. Therefore, parametric statistics are tricky while dealing with this issue. Rank correlations help us mitigate this problem to a degree, because it doesn’t require us to verify any assumptions, and have been shown to be more robust for small sample sizes (Mundry and Fischer, 1998) (though we note that Spearman is also sensitive to outliers (Rousselet and Pernet, 2012)). For this reason, we highlight robust correlations as the best solution here (see also our response to reviewer #2

below, who raised some important considerations). We now appreciate that including the Spearman values in the figure have given the wrong impression that this is the best alternative. To avoid this, we have removed the rho values from the figure.

“Robust correlation methods (e.g. bootstrapping, data winsorizing, skipped correlations) should be preferred in most circumstances because they are less sensitive to outliers (Salibian-Barrera and Zamar, 2002). This is because these tests take into consideration the structure of the data (Wilcox, 2016).”

Reviewer #2:– 'Yet, the use of parametric correlations, such as Pearson's r, requires that both variables are normally distributed.'No, it doesn't! All parametric linear models (as far as I understand) require that the error is normally distributed. In the case of a single-sample t-test against a single mean, this is identical to the requirement that the variables themselves are normally distributed. But, for everything else, it is the differences or error or residuals after the model is fit which must be normally-distributed, not the raw data. The authors repeat in their tutorial what I understand to be a very common mis-interpretation, and it would be good for them to make absolutely certain that what they say here is correct, to avoid perpetuating these errors.

We thank the reviewer for this education! We have now revised the text as follows:

“Correlations are an important tool in science in order to assess the magnitude of an association between two variables. Yet, the use of parametric correlations, such as Pearson’s *r* relies on a set of assumptions, which are important to consider as violation of these assumptions may give rise to spurious correlations.”

Here is a tutorial from the R team: https://rcompanion.org/handbook/I_01.html, specifically: "In particular, the tests discussed in this section assume that the distribution of the data are conditionally normal in distribution. That is, the data are normally distributed once the effects of the variables in the model are taken into account. Practically speaking, this means that the residuals from the analysis should be normally distributed. This will usually be assessed with a histogram of residuals, a density plot as shown below, or with a quantile-quantile plot… Be careful not to get confused about this assumption. You may see discussion about how "data" should be normally distributed for parametric tests. This is usually wrong-headed."The authors are correct here that (genuine) outliers can lead to spurious correlations, but the remedy for this is, as they state: a) plot the data, b) run some robustness-checks, and to report all the results with their standard errors and with due caution.One real problem is how do we identify 'genuine' outliers? Perhaps a large sample size is one remedy, so that we have a better coverage of the population? Yet, there will still be cases when clear 'outliers' are genuine observations which obey the law that you are trying to discover. For example, measuring mass vs. body length across the animal kingdom: there will be an awful lot of small animals down the bottom of the scale (e.g., insects), some in the middle (e.g., birds and most mammals), and fewer still at the extremes (e.g., whales or elephants). I would bet that the blue whale follows the same statistical law of mass vs. length as the gnat (with variance away from this model due to shape). A 'spurious' correlation might arise from incomplete sampling of the problem space – if we only sampled insects and whales, we might draw the wrong conclusion and call the correlation between mass and length 'spurious'. Or perhaps the data need log-transforming first?My laboured point here is: if you don't have any independent reason to exclude a particular datapoint (e.g., the participant didn't do the task properly, wasn't wearing their glasses, is not healthy, is not typical; the elephant was stretching its legs), then I think it is dangerous to conclude that the correlation is spurious just because of one 'outlier'. Rather: authors need to present the data, check their assumptions, and speculate that sampling bias or experimenter error has led to this 'outlier'. In general, the authors are correct here: limited sampling of the intended population may make such outliers more likely, and correlations may then be more problematic. But this is all relative, and error can occur in both directions (Type I, Type II). The only solution is to be very careful, both in including and excluding data.

We fully take this point and have revised the section to better reflect it.

– 'when the two variables are not independent' – Repeated measures designs will often have highly-correlated scores between different conditions or time-points in the same participants. Again, is the real issue here 'independent error' (residuals) not 'independent data'?

This issue is now being handled in the new item ‘circular analysis’. We took care to avoid this statistical misconception.

– This is the 'regression towards the mean' error that I discussed in Holmes (2007, 2009), yet this topic is only an "Honorable mention" here! I would suggest all these 'circular analyses' and 'double-dips' (i.e., both are experimenter-created dependencies in the data) could be in their own section (after dealing with the below comments, in which I suggest removing point 6 entirely).

We have followed this suggestion and added a new section (Section 6).

– Figure 2A – I would bet that the red point is not an outlier here.

We have revised the figure legend to better reflect the procedure applied here.

“From A to C, the distance between the main population (*N*=20, black circles) and the red circle was systematically varied until it became a formal outlier (C).”

– 'they can run some basic simulations' – I agree 100% (Holmes 2007, 2009), but this sentence will have, in my view, about 95% of your target audience hiding under the bedcovers in fear of programming. How does someone who is not sufficiently well-trained to spot these problems in the first place go about 'running some simulations'? Can the authors point to an online tool or tutorial that helps?

This suggestion has now been moved to circular analysis, and re-phrased as follows:

“If suitable, the reviewer could ask the authors to run a simulation to demonstrate that the result of interest is not tied to the noise distribution and the selection criteria.”

5) Use of underpowered groupsReviewer #1:– Experiments with small samples sizes are quite often small for very good reasons, not always but often. We should not recommend that scientists don't do small experiments – sometimes there is no option – but we should tell them not to report inferential statistics. Particularly if the study does not have an a priori power calculation. Not sure that Figure 3 adds much here.

Thank you for raising this important point. It is certainly true that in some cases (e.g. in research on animals and non-human primates in particular) there is very good reason to limit data collection, and in the revised manuscript we included some solutions to address these cases. But this is the exception rather than the rule. Particularly, in our field of cognitive neuroscience the literature clearly shows that we are often underpowered for bad reasons (Higginson and Munafo, 2016). As a researcher who works with rare patient groups and using expensive neuroimaging techniques, TRM suggests that in many instances this becomes a matter of priority (spending a lot more time and resources to collect a little more data). As a researcher who works with healthy participants using inexpensive techniques, it is JJOdX’s view that there seems to be little reason why people opt to publish studies on very small sample sizes (n<15). We therefore believe that as a community we should raise the bar. We therefore kept the strong emphasis on scrutinising statistical power.

“In situations where sample size may be inherently limited (e.g. research with rare clinical populations or non-human primates), efforts should be made to provide replications (both within and between cases) and to include sufficient controls (e.g. to establish confidence intervals). Some statistical solutions are offered for assessing case studies (e.g., the Crawford *t*-test; (Corballis, 2009)).”

Figure 3 has been removed, as per the reviewer and editor’s request.

Reviewer #2:– 'In frequentist statistics in which a significance threshold of α=.05 is used, 5% of all statistical tests will yield a significant result even in the absence of an actual effect (false positives; Type I error)' – I think the authors need to clarify this a bit more, to, e.g.: "Assuming that the null is true, then randomly- and independently-sampled data from a normal distribution with a mean of zero will yield a sample that, when tested against a mean of zero, has a p-value below or equal to.05 approximately 5% of the time." The word 'even' in their claim here is unhelpful – the stats explicitly assume that the null is true (it is never actually true!)

Since we removed this figure, the text here has been simplified:

“Critically, the larger correlation is not a result of there being a stronger relationship between the two variables, it is simply because the overestimation of the actual correlation coefficient (here, *r*=0) will always be larger with a small sample size. For instance, when sampling two uncorrelated variables with *N*=15, simulated false-positive correlations roughly range between |0.5-0.75| whereas when sampling the same uncorrelated variables with *N*=100 yields false-positive correlations in the range |0.2-0.25| (Code available at https://github.com/jjodx/InferentialMistakes).”

– 'Given that these two variables are random, there should be no significant correlations' – See previous point. There will be 5% 'significant' correlations.

See above.

– 'falsely significant' – I don't like this phrase. It seems contradictory. I know what they mean (something like: 'using the standard α criterion, most researchers would conclude that there is a positive correlation in the population when in fact there isn't').

See above.

– 'the experiment is underpowered' – But there is no effect in the simulated population. There can therefore be no sample size sufficient to find this effect. This cannot, therefore, be 'underpowered'. Revise.

See above.

– '<' should this, in fact, be '≤'? Same throughout the manuscript.

The text has been modified as suggested.

- 'Designs with a small sample size are also more susceptible to Type II errors' – Why? Type II error is a non-linear function of the sample size and the real effect size. Knowing the Type II error requires that you know the population distribution, which is almost never the case (and not required) in the kinds of parametric null-hypothesis tests that the authors are discussing here.

With small samples, it becomes simply more difficult to detect an effect because the power is low. This has now been clarified in the text as follows:

“Designs with a small sample size are also more susceptible to missing an effect that exists in the data (Type II error). For a given effect size (e.g., the difference between two groups), the chances are greater for detecting the effect with a larger sample size (this likelihood is referred to as statistical power). Hence, with large samples, you reduced the likelihood of not detecting an effect when one is actually present.”

– 'based on a limited number of participants' – I would remove this as I don't think this it is justifiable. I think all effects (especially surprising ones) from a single experiment should be taken with the same degree of caution, regardless of their sample size (who sets the criterion in any case?). The statistics deal with the problem of sample size. Statistics can be biased as a function of sample size, of course, and some come with corrections (e.g., Hedges G instead of Cohen's d) but if you expect a large effect (e.g., removing striate cortex will impair vision), then I see nothing wrong with doing the absolute minimum of testing on your subjects to establish that effect. It would be unethical to remove 30 monkeys' visual cortices when 2 are sufficient to test the hypothesis.

We agree that this is experiment-specific and we should therefore be more nuanced in our recommendation.

“In situations where sample size may be inherently limited (e.g. research with rare clinical populations or non-human primates), efforts should be made to provide replications (both within and between cases) and to include sufficient controls (e.g. to establish confidence intervals). Some statistical solutions are offered for assessing case studies (e.g., the Crawford *t*-test; (Corballis, 2009)).”

– 'that were not replicated' – Yes, we should be especially skeptical if a second well-powered experiment failed to replicate the first, but I would only be normally skeptical of a single-experiment finding, regardless of its sample size. I think the authors' main point is that small studies will achieve significance only with large effects. True. But some large effects are real, so given a single particular result, how do you know? We must be allowed to search for large statistical effects. Collecting converging and independent evidences should be sought in all investigations, not just in those researchers looking for large effects: Smith and Little (2018).

We fully take this point (which also relates to reviewer #1’scomments above). We have now revised the text to reflect these considerations more carefully:

“How to detect it: Reviewers should critically examine the sample size used in a paper and, judge whether the sample size is sufficient. Extraordinary claims based on a limited number of participants should be flagged in particular.

Solutions for researchers: A single effect size or a single *p*-value from a small sample is of limited value and reviewers can refer the researchers to Button et al., 2013 to make this point. The researchers should either present evidence that they have been sufficiently powered to detect the effect to begin with, such as through the presentation of an a priori statistical power analysis, or perform a replication of their study. The challenge with power calculations is that these should be based on an a priori calculation of effect size from an independent dataset, and these are difficult to assess in a review. Bayesian statistics offer opportunities to determine the power for identifying an effect post hoc (Kruschke, 2011).”

6) Using parametric statistics with small sample sizes– Sorry but I cannot agree with much of this section. See my previous comments about normality assumptions. Tests of normality are really not very useful in most circumstances. In small samples, I agree with authors, they are useless. However, in large samples they will always reject with probability of very near one.– In general, decisions about normality and whether to use parametric or non-parametric methods should be based mainly on scientific principle. For instance, if I collect data on the heights of 10 people, I report a median and IQR, but if I collect data on 50 people a mean and SD? No, that is clearly wrong. I believe that heights are approximately distributed, based on the way it is measured, my own experience and the experience of others (irrespective of what a test of normality tells me!), so I should summarise data on that basis in the appropriate way by a mean and SD.– This sort of mechanical/automated approach to the implementation of statistical methods is strongly discouraged by the majority of statisticians. This is analogous to the widespread adherence to the (mis-) interpretation of p-values that has been so widely criticized by among others the American Statistical Association. In its guidance for the use of statistical tests for such decisions and the role of p-values it makes clear that "Scientific conclusions and business or policy decisions should not be based only on whether a p-value passes a specific threshold" and "No single index [the p-value] should substitute for scientific reasoning".– Scientific reasoning and precedent should be used to make decisions about how to appropriately analyse data, not arbitrary ad hoc (data dependent) statistical tests.Reviewer #2:– Section 6 is incorrect and should be removed. In part, due to the same problem as point #4 (normal distribution of error), but what the authors seem to be arguing for (unwittingly) is an abandonment of parametric tests, not for their use only with 'large' samples, as explained below:Parametric statistics were designed for small samples. If in any doubt about this, please check out Fisher (1925), available in full here: https://psychclassics.yorku.ca/Fisher/Methods/index.htmIn parametric statistics, from a sample of data we extract parameters, for example the mean and the SD (e.g., of the difference between means of two conditions). In our statistical model we compare those parameters to a normal distribution in order to make probabilistic inferences using the theoretical distribution.When sample size is small (say <30), we do not refer our parameters directly to the normal distribution. Rather, we refer them to the t, F, Chi-Square, Poisson, Binomial, or any other appropriate statistical distribution. These distributions were originally created by sampling small numbers of datapoints, and seeing how they behaved. These distributions are all skewed in interesting ways, but as N increases, they tend to approximate the normal distribution.The (in my view) often-mistaken 'rule of thumb' that you need at least 30 participants to do parametric statistics is wrong. It is in my understanding,opposite this. As N increases, the t-, F, Binomial, Chi-square, and Poisson distributions converge closer and closer to the normal distribution. So, when N=30, rather than using the t-test, you can just use the Z-test (i.e., essentially ignoring sample size). The critical t-value for 1 degree of freedom (N=2) at α=.05 is 6.31 (i.e., 6.31 standard errors of your sample mean difference away from zero). As N increases to infinity, the critical value converges to 1.645. At N=30, the critical t-value is 1.7, which is arguably close-enough to the population Z-score (1.645) that the t-distribution can be abandoned (i.e., only sample size is relevant for calculating the SE, df is not needed) and that the Z-distribution can be used instead. Small samples are already 'punished' via the df, by requiring much larger effect sizes to pass arbitrary statistical thresholds.My understanding is that most statistical tests are designed for small samples. Non-parametric tests are for non-interval and non-ratio data (categorical, ordinal), or for interval/ratio data with populations for which no reasonable assumptions can be made (e.g., with large inexplicable outliers). Bootstrapping or other non-parametric statistical methods can be useful to check whether small samples are indeed sufficiently normal to use parametric tests (e.g., from Makin et al., 2009, where Ns=6-11: "In every case, this bootstrapping procedure supported the inferences derived from the t tests, so we report only the standard parametric tests in this manuscript.").As the authors correctly note, small samples come with biases (e.g., effect sizes are larger for significant effects), but this doesnot invalidate the use of parametric tests. The authors cite Kar and Ramalingam (2013) in support of their claim, yet from that paper's conclusion: "Hence, there is no such thing as a magic number when it comes to sample size calculations and arbitrary numbers such as 30 must not be considered as adequate."– 'though it is well agreed that you shouldn't use a parametric test with N<10 (Fagerland, 2012)' – The authors present no evidence for this 'well agreed' rule. I disagree with it, for example, as I believe would the statisticians who invented parametric tests. The article cited by Fagerlund, (2012) was specifically looking at skewed distributions (gamma and log-normal) of the underlying population parameters, and made the same common mistake about normal distribution of the data (should be: error/residuals). Yes, with data producing skewed error distributions, transformation or non-parametric tests are required, or a larger sample, and the central limit theorem can be relied upon. It is absolutely fine to use parametric tests under reasonable assumptions and with reasonable caution with N as low as 2 (Fisher, 1925).– 'Nonparametric tests… are less sensitive to outliers' – Bootstrapping is a kind of non-parametric test. Such tests may highlight outliers by revealing a multi-modal distribution of summary statistics, but they are just as 'sensitive' to outliers. Inspecting the distributions and checking assumptions is the correct approach.

This section has now been removed, and as such we will not provide point-by-point responses to the reviewers’ comments.

7) Flexibility of analysis: p-hackingReviewer #1:– The authors advice on how to detect p-hacking is well-meaning but naïve.In truth, only by pre-registering and providing detailed analysis plans, such as we do in clinical trials, can we ever hope to stop p-hacking. Almost impossible for a reviewer to make much assessment of this, unless they have a study protocol available against which to assess the reporting adherence.

We agree – this is why we started the “how to detect it” section with the following disclaimer: “Flexibility of analysis is difficult to detect because researchers rarely disclose all the necessary information”. Nevertheless, in an effort to be proactive, we believe reviewers should challenge the authors when the offered analysis is not straightforward/well justified/consistent with previous publications. In the revised manuscript we highlight the usefulness of pre-registered protocols in helping detect p-hanking and re-emphasise the difficulty of detecting it in the “How to detect it” section.

8) Failing to correct for multiple comparisonsReviewer #1:– This is a tricky one. The truth is that there is no great consensus amongst statisticians as to the best correction method to use. It is very application area dependent and there are many who would simply disagree on principle that correcting for multiple testing makes any sense (e.g. Rothman, 1990, Epidemiology).I would draw a distinction between exploratory and confirmatory analyses, and make differing recommendations dependent on the aims of the study. We may be more or less worried about false negatives and false positives in these settings.This topic is complex and probably beyond the knowledge of most (non-expert) reviewers and beyond the scope of this article.

Agreed. We now highlight this important distinction, and the need to be more thoughtful about the circumstances leading for correction for multiple comparisons (or lack thereof).

“When researchers explore task effects, they often explore the effect of multiple task conditions on multiple variables (behavioural outcomes, questionnaire items, etc.), sometimes with an underdetermined a priori hypothesis. This practice is termed exploratory analysis, as opposed to confirmatory analysis, which by definition is more restrictive. When performed with frequentist statistics, conducting multiple comparisons during exploratory analysis can have profound consequences for the interpretation of significant findings.”

And:

“Therefore, when conducting exploratory analyses with a large set of variables (genes, fMRI voxels, EEG timepoints), it is simply unacceptable for the researchers to interpret results that have not survived correction for multiple comparisons, without clear justification. Even if the researchers offer a rough prediction (e.g. that the effect should be observed in a specific brain area or at an approximate latency), if this prediction could be tested over multiple independent comparisons, it requires correction for multiple comparisons.”

Reviewer #2:– 'it is simply unacceptable for the researchers to interpret results that have not survived correction for multiple comparisons' – Even if hypothesised? Perhaps 'exploratory' needs to be added here. I disagree that if an effect 'could' be tested using different comparisons, then corrections for multiple comparisons are required. A reviewer could just say: “well, you could have done this on all the individual blocks of data rather than the subject averages, so you need to correct…” This would be r-hacking (reviewer hacking), and needs to be discouraged. Explicit, limited, pre-registered hypothesis-testing should be encouraged. Exploratory testing is fine, but should be acknowledged and α corrected.

Agreed. The distinction from confirmatory analysis has been now added (see above), and the sentence changed as suggested. We also highlight that exploratory testing is absolutely appropriate, but should be acknowledged and α corrected.

9) Over-interpreting non-significant resultsReviewer #1:– This is a common error – so common in fact that it is hard to believe anything we suggest will make much difference! The suggestions of the authors are reasonable, but a bit wishy-washy.

Agreed – this is a hugely common issue, and we strongly believe that this could be successfully diagnosed and remedied as part of the peer-review process. As such, we are keen to highlight it. We think our suggestion for either justifying the null effect using specialised statistics or adjusting the interpretation of the test is concrete. But if the reviewer has further suggestions, we would of course be happy to add these.

Reviewer #2:– 'non-significant effects could literally mean anything' – So could significant effects. All the problems listed apply equally to significant effects: true positives, over-powered small effects (e.g., much smaller than the meaningful effect that a theory predicts), or ambiguous effects. There is nothing special about the α value, as the authors note.

Yes, we acknowledge that there is a greater general problem at hand. Here we are referring to the more specific problem where a non-significant p-value does not distinguish between the lack of an effect due to it not being there (contradictory evidence to the hypothesis), or due to insensitivity of the data to the hypothesis (e.g. due to lack of statistical power, or inappropriate experimental design). As such, it is uninterpretable under *any* current framework, and is therefore erroneous, as highlighted by reviewer #1 above. We agree that it’s important to point at the (bigger) elephant in the room – the lack of interpretability of the p value, which we dedicate the final part of the manuscript towards. We hope that this provides a balance between the immediate problem of the interpretation of a null result and the limited interpretability of the p value more broadly. We changed our title of the section to make this distinction clearer and slightly rephrased our text to better reflect our meaning:

“Much has been written about the arbitrariness of this threshold (Wasserstein et al., 2019) and alternatives have been proposed (e.g.,.005; (Benjamin et al., 2018; Colquhoun, 2014; Lakens et al., 2018). […] In simple words – non-significant effects could literally mean very different things – a true null result, an underpowered genuine effect, or an ambiguous effect (see (Altman and Bland, 1995) for an example).”

– 'Otherwise, researchers should not over-interpret non-significant results and only describe them as non-significant.' – So, p=.049 is "significant" and can be interpreted, and p=.051 is "non-significant" and should 'not be over-interpreted'. I think we can do better than this. What rules of thumb do the authors offer to get around this linguistic threshold? My view would be that if there is any doubt in a particular result, then plot the data, check assumptions, run simulations, replicate the experiment with increased power, seek converging evidence, do a systematic review and meta-analysis, present the work at conferences, ask reviewers… Being told to stick rigorously to the 'significant/non-significant' dichotomy is not going to improve the readers' statistical inferences.

We fully agree with this point, and tried to reflect it in our commentary. First, in the specific section we encourage a discussion of the effect size, and the nature of the evidence. We also highlight the problematic notion that the p-value associated with a given statistical test represents its actual error rate (see our Discussion section). Finally, we ask that non-significant results are not over-stated, this is not to say that trends toward significance are ignored!

“This confound also means that sometimes researchers might ignore a result that didn’t meet the *p*≤0.05 threshold, assuming it is meaningless when in fact it provides sufficient evidence against the hypothesis or at least preliminary evidence that requires further attention.”

10) Correlation and causationReviewer #1:– My sense is that scientists generally have a good understanding of this issue. Not really a statistical issue per se, more about using cautious language when reporting.

Agreed. This is not a statistical issue but rather an inferential mistake. This is why we emphasise in our Title/Introduction that we are not restricting the list to purely statistical issues.

– Figure 4 does not show a 'correlation' but two time series data; the right y-axis looks like negative numbers because of the axis ticks; the blue dataset has auto-correlation (mostly the same people eating margarine across different years) but the red does not (mostly different people getting divorced). Since the authors did not create this figure, I suggest removing it from their tutorial. I also suggest they cite primary research data, rather than secondary websites (particularly when that website labels a correlation of r[without degrees of freedom]=.9926 as a '99.26%' correlation. It's not).

This figure has been now removed.

Honourable mentionsReviewer #1:– I really don't think section adds much, just a list of terms with little or no more explanation. I would advise deleting.

We take the reviewer’s point. With the inclusion of the circular analysis many of these issues have been now explicitly discussed. The other issues have been incorporated, if relevant, throughout the manuscript.

Reviewer #2:– 'Impossibly high correlations' – Replace with 'effect sizes'?

Revised (and moved to section 6).

ConclusionsReviewer #2:– Interesting as it is, I don't see why we need a discussion of NHST and p-values as the conclusion. Seems a bit off-topic. A summary of main issues and overlap of the common mistakes and importance would be much more useful. And some recognition of the importance of talking to your statistical colleagues. Most of the issues discussed here are very common issues that all (any) statisticians will be well placed to help with. Whether at study development, writing or reviewing stages of research.

We agree with both points. We have now changed the name of this final section to “Final remarks” to better reflect the content conveyed. With respect to statistical advice, while we fully agree that this could help mitigate all of the issues raised here, we are not confident that this is constructive advice. In our community, statistical advice is not sought out as standard practice. As such, there are not good available resources to seek out such advice. Instead, by suggesting an intuitive explanation of the issues at hand and how to resolve them, we provide a new resource to our community. Indeed, we don’t think advanced statistical training is necessary to avoid these mainstream issues.

[Editors' note: further revisions were requested prior to acceptance, as described below.]

Thank you for submitting the revised version of "Ten common inferential mistakes to watch out for when writing or reviewing a manuscript" for consideration by eLife. This version has been seen by the two reviewers who reviewed the original version (Nick Parsons; Nick Holmes), and their comments are below. It should be straightforward to address these comments, so I would like to invite you to submit a second revised version that addresses these comments.Reviewer #1:Summary:The authors have clearly worked tremendously hard to make changes to the manuscript. It is now quite difficult to read, given all the additions and deletions, so will need a good proof-read to make sure it still makes good sense and scans properly. I make a number of responses to these changes:

We can assure the reviewer (and editor) that the manuscript had been proof-read by a colleague who is also a native English speaker prior to submission.

Essential revisions:I still think it is a strange argument to state initially that this paper is motivated by "…ineffective experimental design, inappropriate statistical analysis, and/or flawed reasoning, appearing in published neuroscience papers…", and then to say a little later that all the issues highlighted are "…applicable across a range of scientific disciplines that use statistics to assess findings…". The latter is true – the issues highlighted are very familiar to most applied statisticians who work in science. If there are particular issues that relate to neuroscience – which I imagine there may well be – then that does not come across at all in this manuscript. For me, I think this manuscript would have worked much better if 'neuroscience' had been added to the title and examples and issues specific to neuroscience had been used throughout. Overall, I do not think that this manuscript really delivers on its aim of responding to and talking directly to neuroscience readers. The issues picked-up are the usual suspects; the kind of issues that statistical reviewers and applied statisticians are very familiar with. Nothing wrong with that, per se, but maybe a missed opportunity to do something more impactful. For instance, a survey of the published literature, and description of common reporting and analysis errors would have been an excellent way of motivating this manuscript. Not seriously suggesting that this be done now. But this has proved to be a highly effective way of making real changes to the research culture and the way research is done and reported in other disciplines.

The current Introduction is the result of a compromise between our own intentions (as stated in our original draft) and the reviewer’s recommendations in the previous round of revisions. As stated in our Introduction, our analysis of these 10 common mistakes is based on our own personal experience as manuscript readers, which is based on multiple sub-disciplines affiliated with the neurosciences. From the extensive reading we have carried out while writing this commentary (as exemplified in our reference list) we have since learned that, unsurprisingly, these are very common mistakes across scientific disciplines. Therefore, we do not wish to argue in any way that these are specific issues to neuroscience, we are simply arguing that these are indeed common to neuroscience. This sentiment has been conveyed in the second paragraph of the Introduction.

To exemplify some of the mistakes, we tried to use broad examples, given the massive diversity in practice across the neurosciences. But I can assure the reviewer that we have focused the manuscript around our field – with the exception of the Nobel laureates (which is quite striking!), the examples are all related to neuroscience. We have no objection to adding ‘neuroscience’ to the title, although as highlighted by the reviewer – it would be good avoid these mistakes when writing *any* scientific manuscript, so we’re not sure this changed title will make sense. I leave the decision to the editor.

With respect to the reviewer’s comment that these issues are ‘the usual suspects’ – we absolutely agree! Because these are such common issues, many previous attempts have been made to address them. But we wish to highlight two aspects that dissociate our effort from previous attempts to improve the research culture. First, previous comments/analyses (especially those including surveys of published literature) tend to focus on one key issue, or several related issues. By concisely summarising a range of common issues in one list, we hope the relative breadth of our commentary will provide a yet non-existing handy tool to assist our community, and in particular early career researchers who are looking for guidance while learning how to review manuscripts. Second, we wish to highlight the online tool that we have developed to accompany this commentary. The tool is designed not only to survey which issues are most prominent but also invite the community to offer alternative solutions to ours, thus promoting a constructive discussion on how to change our research culture. In the revised manuscript, we further emphasise these two important aspects in the Introduction:

“Our list is by no means comprehensive. […] We also hope that by critically considering these issues and our potential solutions, researchers will grow more vigilant about repeating these mistakes in their own manuscripts.”

0) IntroductionOK. This is clearly a matter of perspective. I get asked to review papers, with a specific request asking me to look at statistical issues, quite frequently. These often come after requests from non-statistical reviewers who suggest the editor gets a statistician to look at the manuscript.

Agreed – it is indeed a matter of perspective.

2) Interpreting comparisons between two effects without directly comparing themNot really convinced that this is type of "…erroneous inference.…" is "…very common…" in published papers. Almost the first thing we teach in statistics is how to compare group A to group B, using an appropriate statistical test. The error that the authors highlight here feels much more complicated than this; to compare the mean response in two groups, would I really test each against the null hypothesis that the mean is 0, and then conclude if I reject for one group, then I can infer that this group is 'statistically significantly' different to the other group? If this is done, then it is done to deliberately (maliciously) mislead the reader.

This is such a common problem that a previous (survey) paper dedicated to highlighting it was published (Nieuwenhuis et al., 2011; since then cited over 550 times). Authors will often identify an effect of interest (let’s say in group A), they will then examine the effect in a control group (group B), and will report that the effect was not significant for group B. They will therefore infer that their result is significant, after “ruling out” or accounting for the confounds related to the control group B. This paper has been cited over 570 times. Anecdotally, we have asked Chris Baker to comment on our manuscript (previous editor of The Journal of Neuroscience – our main society journal, and the senior author of the famous double dipping paper – cited 1870 times). He highlighted this issue as the single most common in his view. We have re-worded this section to better explain what the problem is, we hope that it is clearer now.

4) Spurious correlationsThe point I was trying to make about adding confidence intervals seems to have been misunderstood. I would make the general point that, if possible (which it generally is), all point estimates of quantities should be presented with errors; e.g. CIs, range, standard error, bootstrapped intervals etc. The point being that if this had been done for the correlations, then the effect of the outlier on inferences would be much more obvious than it is in a plot that simply presents a point estimate of a correlation as a straight line. The statement that the points in red are clear 'outliers' – presumably because they are a long way from the fitted line – would be much less sustainable as an argument if the line were actually a region of plausible values, given the observed data.

We apologise for misunderstanding the reviewer’s comment. We have now added CI of the individual samples.

I am absolutely not suggesting that data points should be discarded based simply on post-hoc visualisation of the data. The critical point is that scientists need to always question their own data – and not just at the end of a study when all the data have been collected and they no longer remember why one value is far away from all the others. It is perfectly acceptable to discard values, if there is good reason to believe that something has gone wrong or been recorded incorrectly. This happens all the time – e.g. numbers accidently recorded in the wrong units, calibration not done, days and months confused in dates – there is an almost infinite number of ways that data can be 'wrong'. It is perfectly acceptable to change values such as this – I would advise that the raw data remain unchanged and edits are made in a revised dataset, with changes documented and agreed by all in the study, and made available for others to inspect. If we do this, then what do we make of the data-points the authors highlight as 'outliers'? They are either true errors (things went wrong, but we can't find a reason) or then true data-points. In my view, neither case is a good reason to suggest using robust-correlations, if the rest of the data look reasonably normally distributed. Necessary if you believe the latter is true to modify the model being used, or in the former maybe restrict inferences (model fitting) to the region where you have good data, and not include the extreme value(s).It is not correct to believe that for instance using a non-parametric method is a solution (good alternative) to a parametric method, in the setting described here. As I stated in my original comment, because values are converted to ranks you simply move the extreme value in Figure 2C closer to the other values. So really what you are doing is saying that you do not believe the data as recorded are correct in the sense that they can be treated in the way that is implied by the plots; i.e. that they are continuous measures, where the distance between them has some sense of measuring the 'closeness' (e.g. Euclidean distance). So if you use a non-parametric correlation you are really saying that you do not believe the value is 'correct'. Ultimately this goes right back to the basics of designing an experiment and writing a statistical analysis plan (SAP) at the start of a study before data collection begins. In that case you need to make clear what your beliefs are about the metric properties of each study outcome – you cannot simply choose to use a non-parametric analysis after the data are collected because that makes life easier. Sorry to labour the point, but this is important, as it goes to the heart of many of the statistical problems in reporting of science.

When the data are normally distribution, robust correlations give the same answer as a Pearson’s correlation. When the data is not normally distributed, a Pearson correlation can be misleading. Crucially, robust correlations ensure that the reported correlation is not driven by a few points or outliers (as we referenced in our original response). Even if those outlier points are valid, they invalidate the statistical technique with which the relationship is assessed. This point was conveyed in the text as follows:

“But if this true observation is at risk of violating the assumptions of your statistical test, it becomes spurious de facto, and will therefore require a different statistical tool.”

In our opinion, and supported by the references we offer, robust correlations are a good solution which is readily available for minimising spurious correlations (note that reviewer #2, to whom we have also highlighted this suggested solution, was satisfied with this suggestion). But as we stated in the Introduction, our aim is not to dictate the new gold standard in the field for statistical best practice. Instead, we hope to facilitate discussion on how to best resolve these issues under diverse circumstances, as afforded by our online tool. We emphasise that there often exist many alternative solutions for addressing the problems we describe. If the reviewer wishes to propose a key reference conveying their perspective we will be very happy to add it as ‘further reading’.